# GSK-3 signaling in developing cortical neurons is essential for radial migration and dendritic orientation

Meghan Morgan-Smith[1,2]*, Yaohong Wu[1], Xiaoqin Zhu[1], Julia Pringle[1], William D Snider[1,2]*

[1]UNC Neuroscience Center, University of North Carolina, Chapel Hill, United States; [2]Neurobiology Curriculum, University of North Carolina, Chapel Hill, United States

**Abstract** GSK-3 is an essential mediator of several signaling pathways that regulate cortical development. We therefore created conditional mouse mutants lacking both GSK-3α and GSK-3β in newly born cortical excitatory neurons. *Gsk3*-deleted neurons expressing upper layer markers exhibited striking migration failure in all areas of the cortex. Radial migration in hippocampus was similarly affected. In contrast, tangential migration was not grossly impaired after *Gsk3* deletion in interneuron precursors. *Gsk3*-deleted neurons extended axons and developed dendritic arbors. However, the apical dendrite was frequently branched while basal dendrites exhibited abnormal orientation. GSK-3 regulation of migration in neurons was independent of Wnt/β-catenin signaling. Importantly, phosphorylation of the migration mediator, DCX, at ser327, and phosphorylation of the semaphorin signaling mediator, CRMP-2, at Thr514 were markedly decreased. Our data demonstrate that GSK-3 signaling is essential for radial migration and dendritic orientation and suggest that GSK-3 mediates these effects by phosphorylating key microtubule regulatory proteins.

*For correspondence: meghan_morgan@med.unc.edu (MM-S); wsnider@med.unc.edu (WDS)

**Competing interests:** The authors declare that no competing interests exist.

**Reviewing editor**: Freda Miller, The Hospital for Sick Children Research Institute, University of Toronto, Canada

## Introduction

Glycogen synthase kinase (GSK-3) α and β are serine/threonine kinases that act as key downstream regulators in multiple signaling pathways, including Wnt/β-catenin, receptor tyrosine kinase (RTK)/PI3K, and Sonic hedgehog (Shh) (*Kaidanovich-Beilin et al., 2012*). GSK-3s act via mechanisms that include regulation of transcription factors, control of multiple aspects of cellular metabolism, and phosphorylation of cytoskeletal proteins (*Hur and Zhou, 2010*; *Kaidanovich-Beilin and Woodgett, 2011*). Most often, although not invariably, GSK-3s function as negatively acting kinases by inhibiting the functions of substrates at baseline. Inhibition is then relieved via signaling pathways that engage GSK-3 (*Doble and Woodgett, 2003*; *Kaidanovich-Beilin and Woodgett, 2011*). For most GSK-3 substrates, phosphorylation by another kinase near the GSK-3 site ('priming') is required for, or enhances, GSK-3 substrate phosphorylation (*Cohen and Frame, 2001*; *Doble and Woodgett, 2003*; *Kaidanovich-Beilin and Woodgett, 2011*). Priming kinases for GSK-3 substrates include cyclin dependent kinase-5 (cdk5), a kinase that is known to regulate important neurodevelopmental events like radial migration (*Tanaka et al., 2004*; *Cole et al., 2006*; *Li et al., 2006*; *Xie et al., 2006*).

In the nervous system, GSK-3β has long been thought to be a target of lithium used in treatment of bipolar disorder (*Klein and Melton, 1996*; *O'Brien et al., 2004*). Some of the GSK-3β effects related to lithium actions are due to regulation of signaling downstream of dopamine receptors (*Beaulieu et al., 2004*, *2008*; *Urs et al., 2012*). More recently GSK-3 signaling has been implicated in the pathogenesis of schizophrenia (*Emamian et al., 2004*; *Mao et al., 2009*; *Emamian, 2012*). Disrupted in Schizophrenia-1 (DISC1), mutated in some familial cases of schizophrenia, is thought to function in part by modulating GSK-3β effects on progenitor proliferation (*Mao et al., 2009*; *Singh et al., 2011*).

**eLife digest** In the brain, one of the most striking features of the cerebral cortex is that its neurons are organized into different layers that are specifically connected to one another and to other regions of the brain. How newly generated neurons find their appropriate layer during the development of the brain is an important question; and, in humans, when this process goes awry, it can often result in seizures and mental retardation.

An enzyme called GSK-3 regulates several major signaling pathways important to brain development. The GSK-3 enzyme switches other proteins on or off by adding phosphate groups to them.

Morgan-Smith et al. set out to better understand the role of GSK-3 in brain development by deleting the genes for this enzyme specifically in the cerebral cortex of mice. Mice have two genes that encode slightly different forms of the GSK-3 enzyme. Deleting both of these in different groups of neurons during brain development revealed that a major group of neurons need GSK-3 in order to migrate to the correct layer. Specifically, the movement of neurons from where they arise in the central region of the brain to the outermost layer (a process called radial migration) was disrupted when the GSK-3 genes were deleted.

Morgan-Smith et al. further found that cortical neurons without GSK-3 were unable to develop the shape needed to undertake radial migration because they failed to switch from having many branches to having just two main branches. Additional experiments revealed that these abnormalities did not depend on certain signaling pathways, such as the Wnt-signaling pathway or the PI3K signaling pathway that can control GSK-3 activity.

Instead, Morgan-Smith et al. found that two proteins that are normally targeted by the GSK-3 enzyme have fewer phosphate groups than normal in the cortical neurons that did not contain the enzyme: both of these proteins regulate the shape of neurons by interacting with the molecular 'scaffolding' within the cell. The GSK-3 enzyme was already known to modify the activities of many other proteins that affect the migration of cells. Thus, the findings of Morgan-Smith et al. suggest that this enzyme may coordinate many of the mechanisms thought to underlie this process during brain development.

Despite the obvious importance of GSK-3 signaling in pathogenesis and treatment of psychiatric disorders, there are important gaps in information on the role of GSK-3 in the developing brain. It has clearly been established that GSK-3 signaling is a strong regulator of radial progenitor proliferation in the developing cerebral cortex and that these effects are at least partly mediated through β-catenin (*Chenn and Walsh, 2002*; *Kim et al., 2009*). Additionally, a recent study demonstrated an important role for GSK-3 in regulating INP amplification, an effect associated with GSK-3β binding to the scaffolding protein Axin (*Fang et al., 2013*). Thus, critical roles for GSK-3 signaling in processes that control neuron number in the developing telencephalon have been established.

In contrast, functions of GSK-3 in regulating developing cortical neurons are much less clear. Multiple in vitro studies have suggested roles for GSK-3 in regulating neuronal polarity and axon growth and branching (see *Hur and Zhou, 2010* for review). These functions are thought to be mediated via GSK-3 phosphorylation of microtubule-associated proteins (MAPs) including Collapsin response mediator protein-2 (CRMP-2) (*Yoshimura et al., 2005*), Adenomatous polypsis coli (APC) (*Shi et al., 2004*; *Zhou et al., 2004*), Tau (*Stoothoff and Johnson, 2005*), microtubule-associated protein 1B (MAP1B) (*Trivedi et al., 2005*), Doublecortin (DCX) (*Bilimoria et al., 2010*), and subsequent regulation of cytoskeletal dynamics. In general, inhibition of GSK-3β via serine 9 (ser9) and GSK-3α via serine 21 (ser21) phosphorylation and subsequent relief of phosphorylation of downstream targets is thought to be required for the formation of the axon and subsequent axonal growth (*Jiang et al., 2005*; *Hur and Zhou, 2010*). In a similar vein, a study employing in utero electroporation of an activating construct suggested that GSK-3 inhibition was essential for radial-guided cortical neuronal migration downstream of STK11 (LKB1) via a mechanism involving APC (*Asada and Sanada, 2010*). A prediction of this work might be that GSK-3 deletion would enhance axon growth and radial migration. However, to date these effects of GSK-3 on neuronal polarity and migration have not been confirmed with mouse genetic studies. Further, mice with point mutation knockins that prevent

ser9/21 phosphorylation are viable and have not been reported to show defects in neuronal morphology or migration (*McManus et al., 2005*; *Gartner et al., 2006*). Finally, GSK-3 may regulate migration and morphology of cortical neurons via entirely different pathways for example via mediating effects of semaphorin signaling (*Chen et al., 2008*; *Renaud et al., 2008*; *Nakamura et al., 2009*).

We have now assessed GSK-3 functions in newly born cortical neurons using *Neurod6* (*Nex*)-Cre (*Goebbels et al., 2006*) to mediate recombination of *Gsk3* floxed alleles in INPs and newly born excitatory neurons. We demonstrate, surprisingly, that GSK-3 activity is essential for radial neuron migration in all areas of the cortex and in the hippocampus. In contrast, tangential migration is not affected after *Gsk3* deletion in cortical interneuron precursors. Remarkably, the migration effects appear to be independent of Wnt/β-catenin signaling that mediates GSK-3 functions in neuronal progenitors. The few upper layer neurons that reached their normal location exhibited strikingly abnormal orientation of basal dendrites. GSK-3 control of migration and morphology is correlated with the regulation of phosphorylation of DCX on ser327 and CRMP-2 on Thr514. We conclude that GSK-3 is a critical regulator of neuronal migration and morphogenesis and that GSK-3 regulation is mediated by phosphorylation of key cytoskeletal proteins.

## Results

### GSK-3 signaling regulates migration of cortical excitatory neurons

To investigate the function of GSK-3 in developing cortical neurons, we generated $Gsk3a^{-/-}Gsk3b^{loxp/loxp}$: *Neurod6-Cre* mice (*Gsk3:Neurod6*). The *Neurod6-Cre* line induces recombination in intermediate progenitors of the dorsal telencephalon and early postmitotic neurons beginning at approximately embryonic day 11 (E11) (*Goebbels et al., 2006*). Prior studies using crosses with reporter lines indicate that recombination occurs in virtually all excitatory pyramidal neurons in the dorsal telencephalon (*Goebbels et al., 2006*; *Monory et al., 2006*). Western blot analysis of lysates from the whole cortex, shows a 60% decrease in GSK-3β protein at E19.5 as compared with heterozygous litter mates (*Figure 1C*). Remaining GSK-3β protein in the mutants is likely due to lack of recombination in interneurons and developing glia. At E19, *Gsk3:Neurod6* brains are roughly the same size as littermate controls. However, *Gsk3:Neurod6* mice die shortly after birth (P0–P3) for reasons that have not yet been determined.

To explore neuronal functions of GSK-3, we first assessed cortical lamination at E16, a time of rapid neuronal migration along radial glial processes. Deep layer neurons appeared to be normally positioned (*Figure 1—figure supplement 1*). Thus staining with Tbr1, a layer 6 marker, revealed a distinctive band in the deeper layers of the cortex in both controls and *Gsk3:Neurod6* mutants. In contrast, we noted clear abnormalities in the localization of Cux1 expressing neurons that normally populate Layers 2/3 (*Figure 1—figure supplement 1*). A clear band of Cux1 positive neurons has formed in controls by E16. In contrast *Gsk3:Neurod6* mice exhibit a dispersion of Cux1 cells with fewer neurons reaching the outermost layer, even at this early developmental stage.

Dramatic mislocalization of layer 2/3 neurons was apparent by E19.5. Coronal sections through developing somatosensory cortex showed a large population of Cux1 expressing neurons essentially 'stuck' in the intermediate zone and throughout the deeper cortical layers (orange arrows) (*Figure 1A–A'*). Indeed some Cux1-expressing neurons in mutants were observed in the ventricular zone (yellow arrowhead). The migration defect in *Gsk3:Neurod6* mutants was striking along the entire rostrol/caudal axis at E18.5, as observed in parasagittal sections (arrows) (*Figure 1B–B'*). The migration defect was particularly prominent anteriorly, a developmental profile corresponding to the neurogenic gradient of the developing cortex (*Caviness et al., 2009*).

We also generated $Gsk3a^{loxp/loxp}$, $Gsk3b^{loxp/loxp}$: *Neurod6-Cre* mice ($Gsk3^{loxp}$:*Neurod6*) using a $Gsk3a^{loxp/loxp}$ mouse line, which harbors loxp sites flanking exon 2 of *Gsk3a*. Western blot analysis of lysates from the whole cortex at P0, verified an 85% decrease in GSK-3α and a 76% decrease in GSK-3β protein when compared with wild-type littermate controls (*Figure 2B*). At P0 as expected, $Gsk3^{loxp}$:*Neurod6* mice showed the same migration failure of upper layer neurons as *Gsk3:Neurod6* (*Figure 2A*). Inspection and quantification, (*Figure 2A,C*) showed most Cux1+ neurons stuck in deeper layers in mutants, whereas in controls a heavy majority of neurons had already migrated to the most superficial layers. An increase in neurogenesis could in theory account for some of the Cux1+ neurons found in deeper layers, but counts of Cux1+ cells revealed no major difference in numbers between *Gsk3* mutants and controls (control Cux1/total = 32.45% ± 3.74, mutant = 34.31% ± 0.649, p=0.561, unpaired t-test).

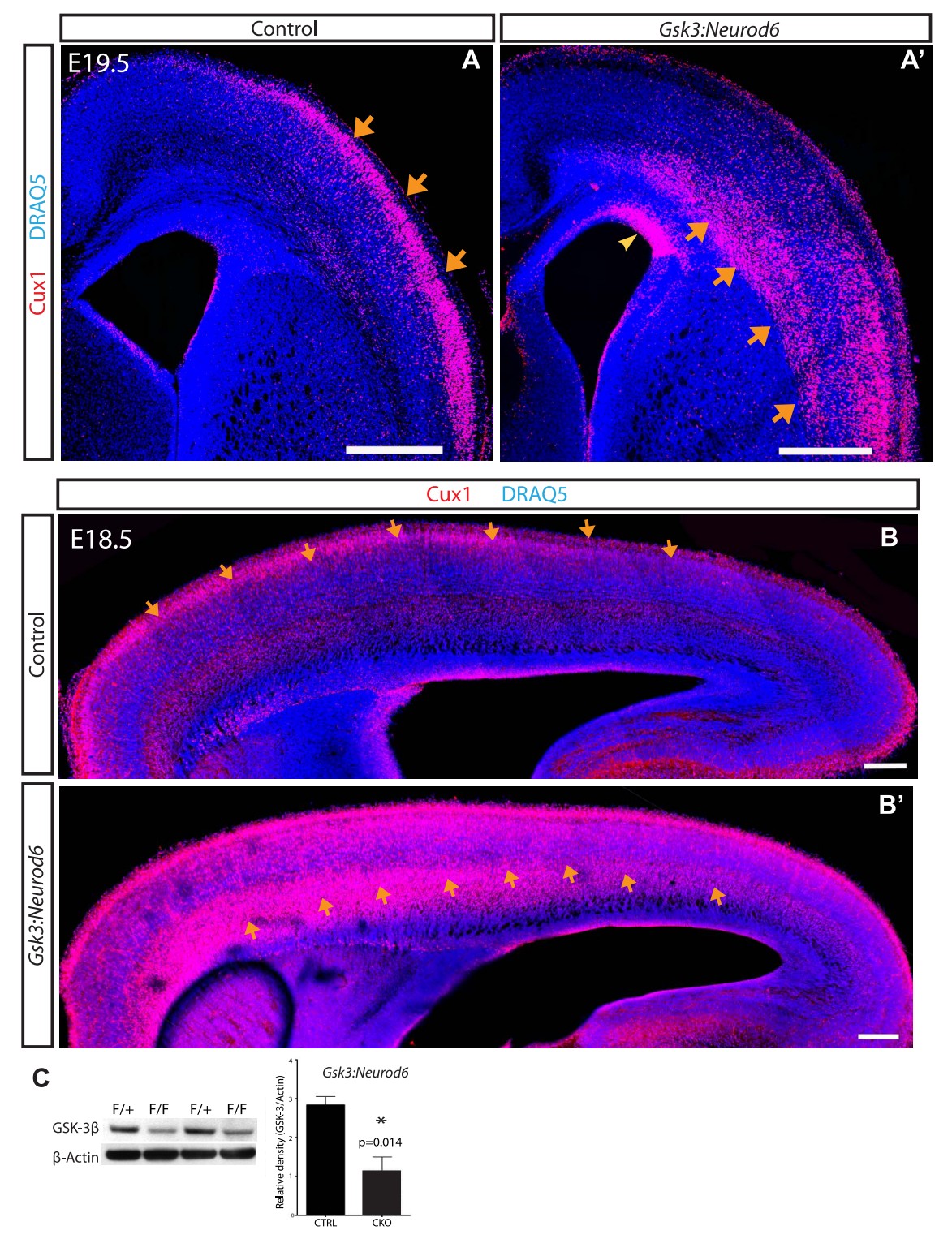

**Figure 1**. GSK-3 signaling is essential for proper lamination of the developing cortex. (**A–A'**) Cux-1 staining (red) in coronal sections from control and *Gsk3:Neurod6* mice at E19.5. Cux-1 neurons are strikingly mislocalized in *Gsk3:Neurod6* mutants (orange arrows) including a small population of neurons that remain in the ventricular zone (yellow arrowhead). Nuclei were counterstained with DRAQ5. Scale bar = 500 μm. (n = 4). (**B–B'**) Cux-1 staining in parasagittal vibratome sections from control and *Gsk3:Neurod6* mutants at E18.5. Cux-1 expressing neurons (arrows) are mislocalized in *Gsk3:Neurod6* mutants and populate the deeper layers of the cortex along the entire rostrol/caudal axis. Scale bar = 200 μm. (**C**) Representative Western blot confirms

*Figure 1. Continued on next page*

*Figure 1. Continued*

strongly reduced GSK-3β protein levels in the E19.5 *Gsk3:Neurod6* cortex compared to heterozygous control (n = 3 het control, n = 3 CKO). Relative Density *p<0.05, unpaired t-test.
The following figure supplements are available for figure 1:

**Figure supplement 1**. Migration defect apparent by E16 after *Gsk3* deletion in cortical excitatory neurons.

A few Cux1 neurons migrated successfully to layer 2/3 (area denoted by yellow lines) in *Gsk3^loxp:Neurod6* mutants (**Figure 2A**). Whether this subset of normally placed neurons did not undergo recombination at an early enough stage for migration to be regulated could not be determined. However, these properly positioned neurons exhibited abnormalities of dendritic orientation (see **Figure 5**).

Interestingly, mice with floxed alpha rather than null alpha alleles survived somewhat longer than the *Gsk3:Neurod6* mice but died after the second postnatal week (P15–P17). At P7, *Gsk3^loxp:Neurod6* mutants exhibited a striking lamination phenotype involving Cux1 neurons (**Figure 2D–D'**). This persistent lamination defect demonstrates that *Gsk3* deletion results in permanent migration failure and does not simply result in a delay.

To further assess functions of GSK-3 in radial migration, we over-expressed *Gsk3b* in newly born cortical neurons using a p*Neurod1* (*Neurod1*) vector (**Guerrier et al., 2009**). Co-electroporation of *Neurod1-Cre* and lox-STOP-lox Ai9 in control embryos at E15 (**Figure 2—figure supplement 1A**) was compared to co-electroporation of *Neurod1-Gsk3b* and *Neurod1*-Egfp in experimental embryos (**Figure 2—figure supplement 1B**). Importantly, over-expression of *Gsk3b* did not inhibit migration as might have been expected from prior studies (**Asada and Sanada, 2010**). In fact, *Gsk3b* over-expression enhanced neuronal migration and resulted in greater than normal numbers of neurons populating the outermost layers of the cortex by E19.5 (**Figure 2—figure supplement 1C**).

## Specificity of GSK-3 signaling for radial migration

Importantly, the migration defect in the developing cortex is specific to excitatory pyramidal neurons. In order to assess interneurons, we used the *Dlx5/6-Cre* (**Stenman et al., 2003**) line to generate conditional mice lacking *Gsk3* in GABAergic interneurons. A robust decrease of GSK-3β protein (84%) was observed in E18 MGE lysates from *Gsk3:DLX5/6-Cre* mice when compared to littermate controls (**Figure 3D,E**). Interneuron migration was monitored using the AI3 reporter line (*Gsk3-Ai3:Dlx*). Surprisingly, in both controls and *Gsk3* mutants, interneurons exhibited robust migration along the two migratory streams (yellow arrows) from the medial ganglionic eminence (MGE) (**Figure 3A–A'**). In *Gsk3* mutants, as in controls, interneurons entered all areas of the cortical plate by E19.5. Quantification is shown in (**Figure 3—figure supplement 1**). These results are not meant to imply that migration of interneurons was normal in every respect as we did not assess migration of specific interneuron subsets.

In order to assess the generality of GSK-3 regulation of radial migration, we assessed migration in developing hippocampus. The hippocampus, like the cortex, is an area where developing neurons migrate along radial glial-like processes (**Nowakowski and Rakic, 1979**; **Eckenhoff and Rakic, 1984**). *Neurod6-Cre* expression is evident in developing hippocampal neurons as early as E14 (**Goebbels et al., 2006**), allowing us to delete *Gsk3* in those cells. Pyramidal neurons generated from the hippocampal primordia undergo migration along radial processes to form CA1/CA3 and the dentate gyrus (DG) (**Altman and Bayer, 1990**). The transcription factor CTIP2 marks neurons in the developing CA1 region (**Figure 3B**). *Gsk3:Neurod6* mice exhibited a striking hippocampal migration defect. CTIP2 expressing neurons did not migrate properly (yellow arrows) and as a consequence CA1 did not fully develop (**Figure 3B–B'**). As a result, CA1-3 and the DG (arrowheads) were disorganized, and the hippocampal sulcus was not well defined (**Figure 3C–C'**). These defects were striking in rostral areas, as shown, although somewhat less pronounced in caudal sections (data not shown). Interestingly, fimbrial axonal projections formed in *Gsk3:Neurod6* mice (**Figure 3B–B'**, orange arrows) demonstrating that even though migration fails, hippocampal neurons were able to polarize and extend appropriately directed axons.

## *Gsk3* deletion delays the multipolar to bipolar transition

To verify that the migration defect was cell autonomous and to visualize morphology of *Gsk3*-deleted neurons, we introduced Cre and EGFP into a subpopulation of developing neurons in *Gsk3a^−/−Gsk3b^loxp/loxp* mice. *Neurod1*-Cre and lox-STOP-lox *lacZ/Egfp* (Z/EG) plasmids were injected

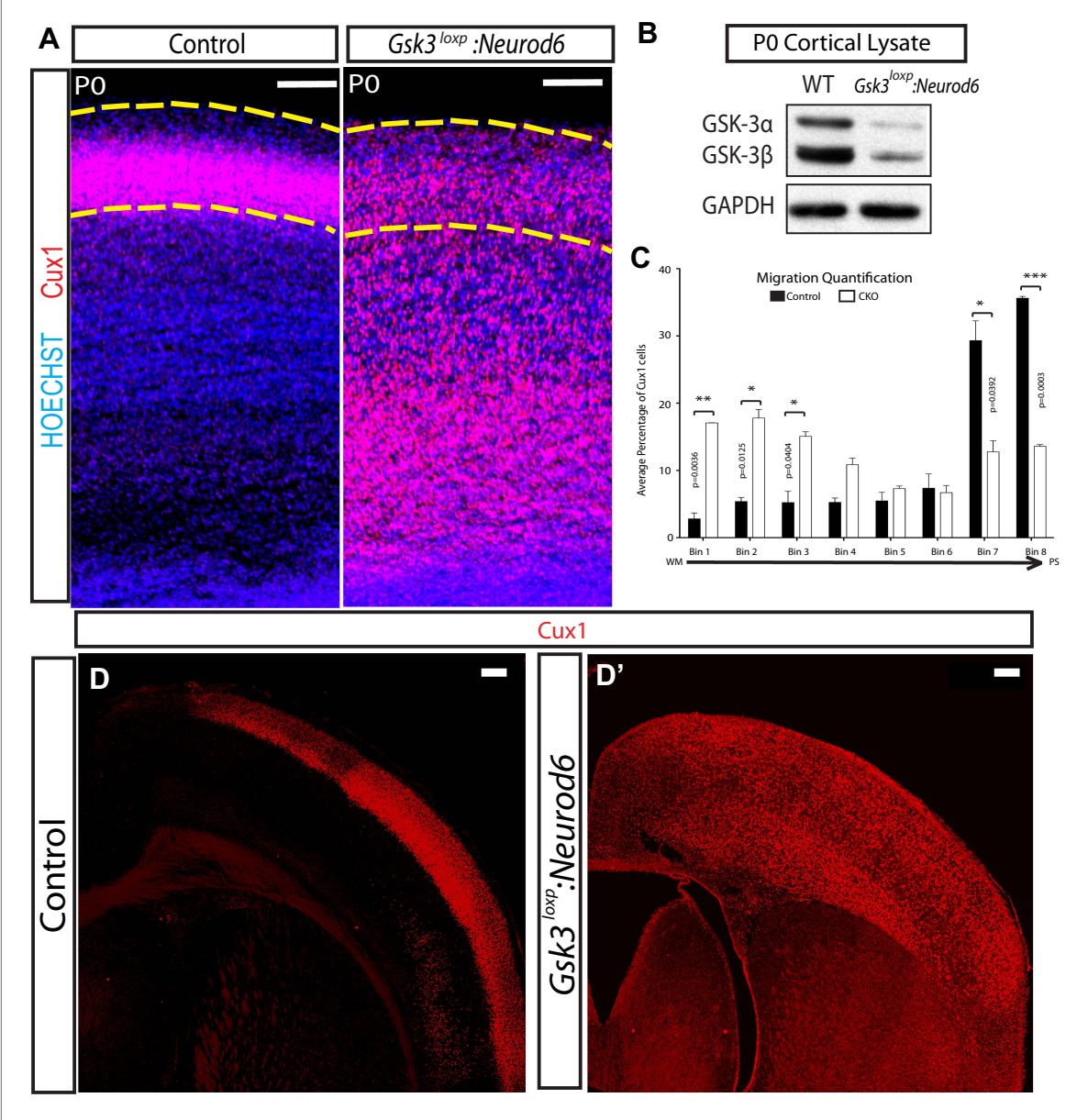

**Figure 2**. Migration defects in *Gsk3*-deleted mice are persistent. (**A**) P0:Cux 1 staining (red) in coronal sections of *Gsk3^loxp:Neurod6* mutants and littermate heterozygote controls. Cux1-expressing neurons are localized to layer2/3 in controls (denoted by yellow dashed lines) while Cux1-expressing neurons are localized throughout the cortical plate in the mutants. (n = 5, scale bar = 100 µm). (**B**) Representative Western blot of P0 cortical lysates confirms strongly reduced GSK-3α and GSK-3β protein levels in *Gsk3^loxp:Neurod6* mutants when compared to *Gsk3a^loxp/loxp Gsk3b^loxp/loxp* controls. GAPDH was probed as a loading control (n = 3 control, n = 3 CKO). (**C**) P0 quantification of control and *Gsk3^loxp:Neurod6* Cux1 neurons using 8 bin analysis spanning white matter (WM) to the pial surface (PS), (n = 2 het control, n = 2 CKO). (**D–D'**) P7: *Gsk3^loxp:Neurod6* mutants stained with Cux1 (red) show persistent altered lamination with Cux1-expressing neurons spread throughout all layers of the cortex. Littermate controls show normal Cux1 distribution in layer 2/3. Scale bar = 200 µm.

The following figure supplements are available for figure 2:

**Figure supplement 1**. Gsk3 overexpression enhances radial migration.

into the ventricles and co-electroporated at E14–15.5. Electroporation at this age targets radial progenitors that generate mainly upper layer neurons. This co-electroporation technique allowed us to visualize individual *Gsk3*-deleted neurons in an otherwise control background. Labeled neurons were imaged at late embryonic and postnatal stages.

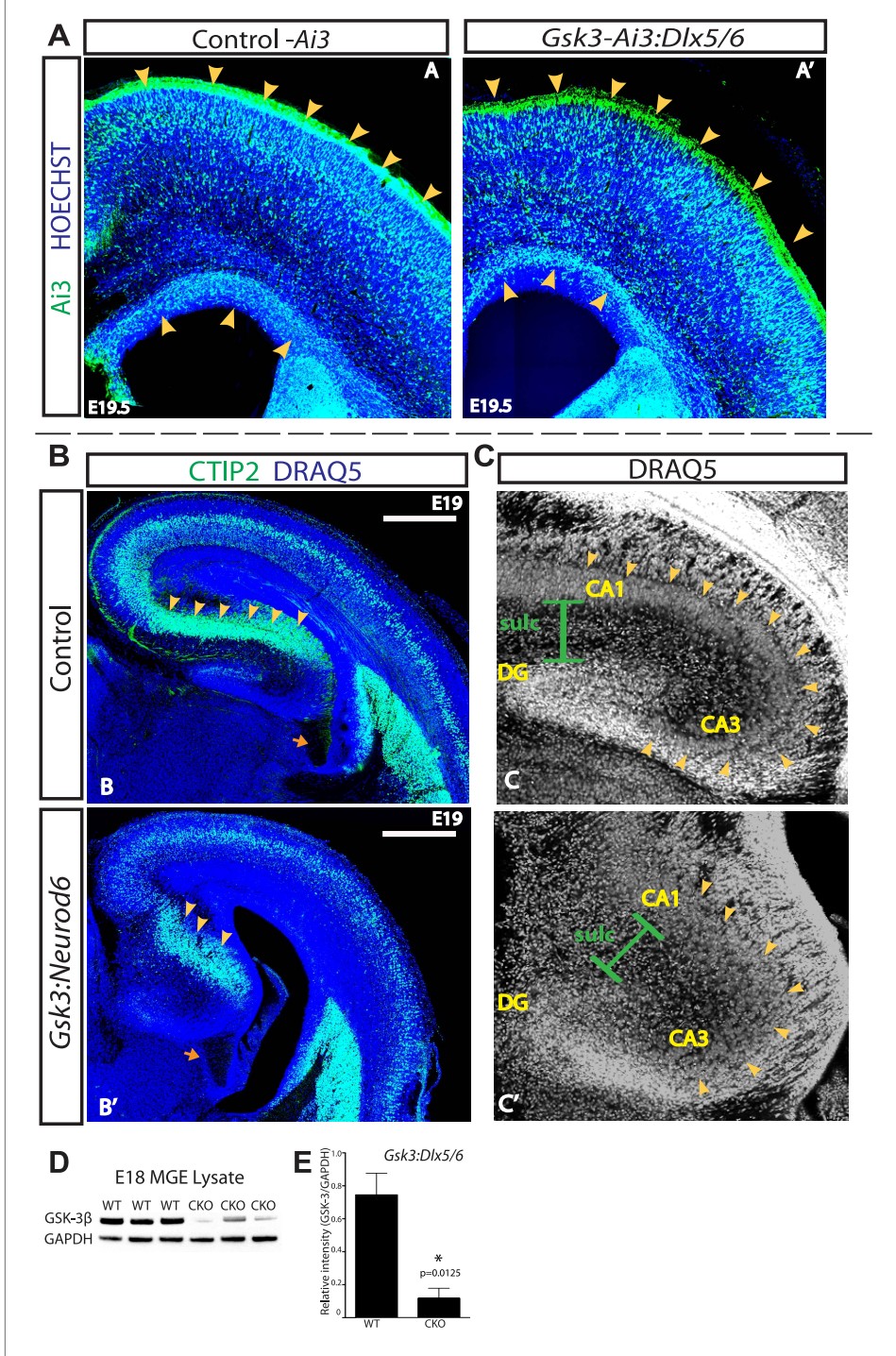

**Figure 3**. GSK-3 signaling is dispensable for tangential migration, but required for radial hippocampal migration. (**A–A'**) E19.5 coronal sections showing EYFP-expressing interneurons in heterozygous control and *Gsk3:Dlx5/6* mutants crossed with the Ai3 reporter line. *Gsk3*-deleted interneurons (green) enter the cortex in two streams in both controls and mutants (arrowheads). Mutants showed no overt migration defect. Nuclei were counterstained with Hoechst. (n = 3). (**B–B'**) E19 coronal sections of control and *Gsk3:Neurod6* mutants showing CTIP2 (green) expressing neurons in the hippocampus. In the *Gsk3:Neurod6* mutants, the pyramidal cell layer (green) does not extend laterally into a compact CA1 region and remains dispersed (yellow arrowheads). Fimbrial axonal projections appear normal in *Gsk3:Neurod6* mutants (orange arrow). Nuclei were counterstained with DRAQ5. *Figure 3. Continued on next page*

*Figure 3. Continued*

Scale bar = 500 μm. (n = 3). (**C–C'**) Higher magnification of hippocampal area shown in (**B**). The *Gsk3:Neurod6* mutants show disrupted cytoarchitecture. In the mutants, DRAQ5-labeled cells are mislocalized and diffuse (arrowheads) and fail to form clearly defined CA1/CA3 regions of the hippocampus. The *Gsk3:Neurod6* mutant mice also lack a clearly defined hippocampal sulcus (green bars) and dentate gyrus (DG). (**D**) Representative Western blot of E18 MGE lysates confirm strongly reduced GSK-3β protein after recombination with *Dlx5/6-Cre*. (**E**) Quantification of protein knockdown in D (n = 3 WT, n = 3 CKO, unpaired t-test).

The following figure supplements are available for figure 3:

**Figure supplement 1**. No apparent migration defect in *Gsk3:Dlx5/6* mice.

Deletion of *Gsk3* in individual neurons phenocopies the migration delay seen in the *Gsk3:Neurod6* mutants. At E19 most control neurons were located in the upper cortical layers as expected (*Figure 4A*). In contrast, most *Gsk3*-deleted neurons had cell somas that were localized to the deeper layers of the cortex and very few were found in upper layers (*Figure 4A'*). Importantly, *Gsk3*-deleted neurons were clearly able to project axons (orange arrows) suggesting that initial polarization had proceeded in the absence of *Gsk3*. Further, most *Gsk3* null neurons in the cortical plate elaborated a long leading process directed toward the pial surface (yellow arrowheads) (*Figure 4A'*). Thus at least the initial stages of dendritic arborization also appeared to proceed in the absence of *Gsk3*.

The GSK-3 migration defect was strikingly persistent into the postnatal period. In mice electroporated at E14–15.5 and analyzed at P10, *Gsk3*-deleted neurons remained in the deeper layers and subcortical white matter (*Figure 4B–B'*). Egfp-expressing neurons co-labeled with Cux1 (red) demonstrating that they had acquired the proper laminar markers but were unable to attain the proper position (*Figure 4C*, arrows). Again, long apical processes that reached the pial surface were elaborated by *Gsk3*-deleted neurons, basal dendrites formed, and axons projecting towards the corpus callosum were evident. Thus, *Gsk3* regulates some critical aspect of migration but not the early stages of axon and dendrite formation.

In quantifying our results, we found that in control animals the majority of electroporated neurons (73.7%) were found in the upper layers of the cortex (yellow dashed lines) by P10 (*Figure 4B–B',D*). In contrast, *Gsk3*-deleted neurons remained in the deeper layers of the cortex and only 23% had reached the upper layer 2/3 by P10 (p=<0.005).

A number of in vitro studies have suggested that *Gsk3* regulates neuronal polarization. It is plausible that some defect or delay in the polarization might account for migration failure. To test this idea, we electroporated *Neurod1-Cre* and Z/EG into the $Gsk3a^{-/-}Gsk3b^{loxp/loxp}$ cortex, plated cortical cells in dissociated culture and assessed stage progression at 3 days in vitro (3DIV). We observed no statistically significant difference in the stage progression between control and *Gsk3*-deleted neurons (*Figure 4—figure supplement 1A*). Further, *Gsk3*-deleted neurons that successfully extended an axon remained highly dynamic (*Figure 4—figure supplement 1B*) and extended and retracted neurites. Thus at least some processes that require complex cytoskeletal regulation proceed normally in the *Gsk3*-deleted neurons.

To assess the cell biological mechanisms of GSK-3 regulation of neuronal migration, we co-electroporated *Neurod1-Cre*;Z/EG or a control pCAG-dsRED construct into the lateral ventricles of control and $Gsk3a^{-/-}Gsk3b^{loxp/loxp}$ mice. This was followed by live imaging of migration ex vivo in a cortical slice preparation (*Hand et al., 2005*) at 3DIV. In controls electroporated with pCAG-dsRED, labeled neurons transitioned from multipolar to bipolar morphology and migrated through the cortical plate over a period of 12 hr, as expected (*Figure 4E*). The progress of individual neurons could readily be tracked and is indicated for three examples by the progress of the colored arrowheads in the three panels. In contrast, most *Gsk3*-deleted neurons failed to translocate through the intermediate zone and remained in a multipolar state in the outer subventricular zone (arrowheads) (*Figure 4E* and *Videos 1, 2*). Further, during the 12 hr of observation most *Gsk3*-deleted neurons did not transition to a bipolar morphology, a step thought to be required for radial-guided migration.

## Abnormal dendritic orientation in *Gsk3* mutants

That fact that multiple cytoskeletal proteins are GSK-3 substrates might suggest that *Gsk3* deletion would have profound effects on dendrite and axonal arborization. To address GSK-3 regulation of

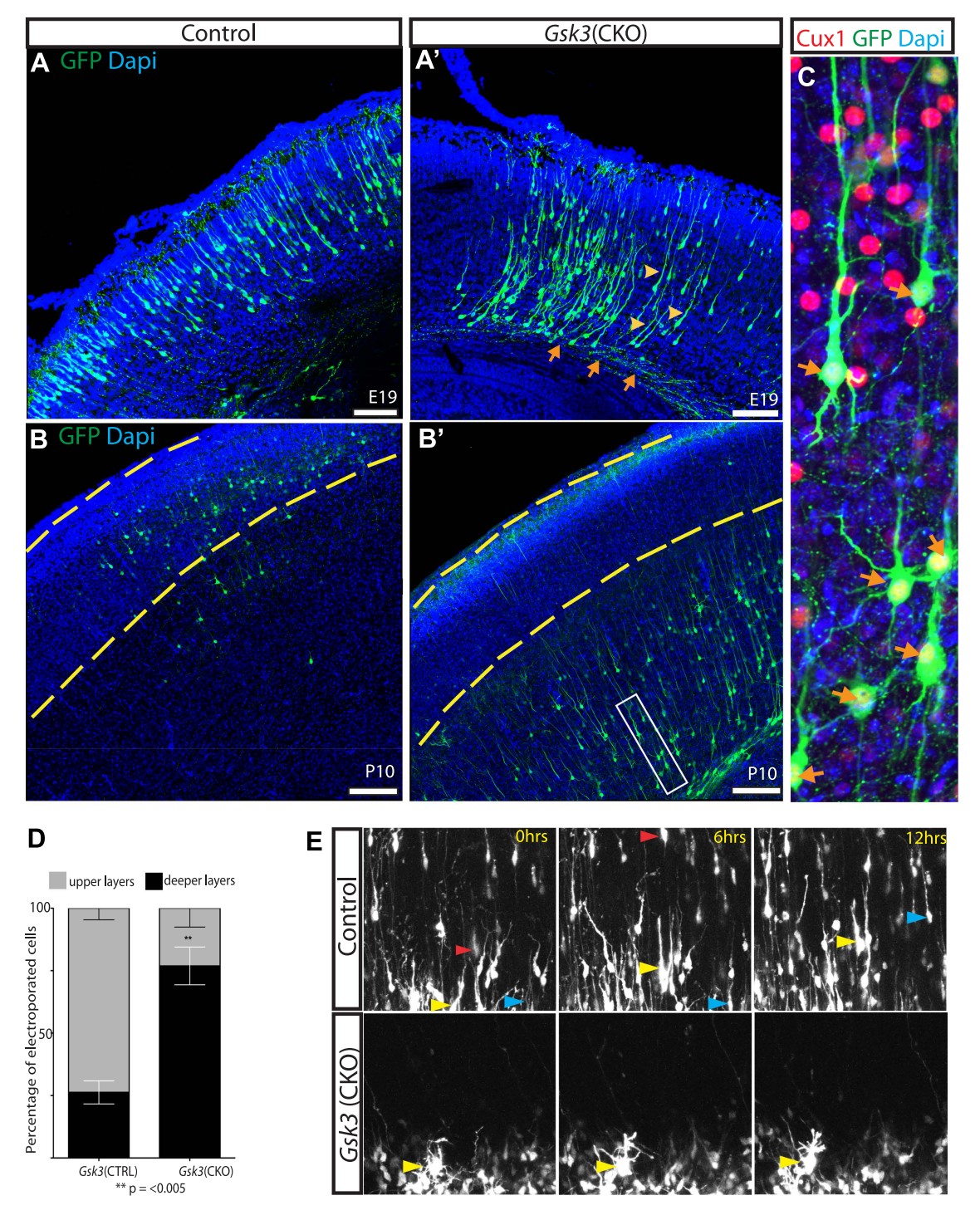

**Figure 4**. GSK-3 deletion delays the multipolar to bipolar transition. (**A–A'**) Representative E19 coronal sections after in utero electroporation at E14.5 with *Neurod1-Cre* and Z/EG plasmids. Electroporated cells were visualized with anti-EGFP (green), and nuclei were stained with DAPI (blue). *Gsk3*-deleted neurons remain in the deeper layers of the cortex but elaborate a long pial-directed process (yellow arrowheads). *Gsk3*-deleted neurons elaborate axons projecting towards the corpus callosum (orange arrows). Scale bar = 200 µm (n = 5, two independent litters). (**B–B'**) Coronal sections at P10 after E14.5 electroporation, as in **A**. *Gsk3*-deleted neurons remain in the deeper layers of the cortical plate and fail to reach layer 2/3 (denoted with yellow bars). Scale bar = 200 µm (n = 3, 2 independent litters). (**C**) Higher magnification of *Gsk3*-deleted neurons in **B'** (box). *Gsk3*-deleted neurons (green) in deeper layers co-label with Cux (red) (orange arrows). Nuclei were stained with Dapi. (**D**) Quantification of control and *Gsk3*-deleted neurons in *Figure 4. Continued on next page*

*Figure 4. Continued*

upper (layer 2–3) vs deeper layers of the cortex at P10. (n = 3, 4209 total neurons counted, 2234 control vs 1975 *Gsk3* deleted) **p=0.003, unpaired t-test. (**E**) *Gsk3* deletion delays the multipolar to bipolar transition. Still images from time-lapse imaging of slice cultures at 3DIV. pCAG-dsRED or *Neurod1-Cre*;Z/EG was injected into the ventricles of *Gsk3a*$^{-/-}$*Gsk3b*$^{loxp/loxp}$ embryos and electroporated at E15. Representative images were taken at time 0, 6, and 12 hr. Control dsRed neurons migrate through the cortical plate (yellow, red, and blue arrows show individual neurons at the different time points). (n = 2 controls). *Gsk3*-deleted neurons fail to migrate through the cortical plate and exhibit persistent multi-polar morphology (yellow arrowheads). (n = 4 mutants).

The following figure supplements are available for figure 4:

**Figure supplement 1**. *Gsk3-deleted* neurons polarize and are highly dynamic.

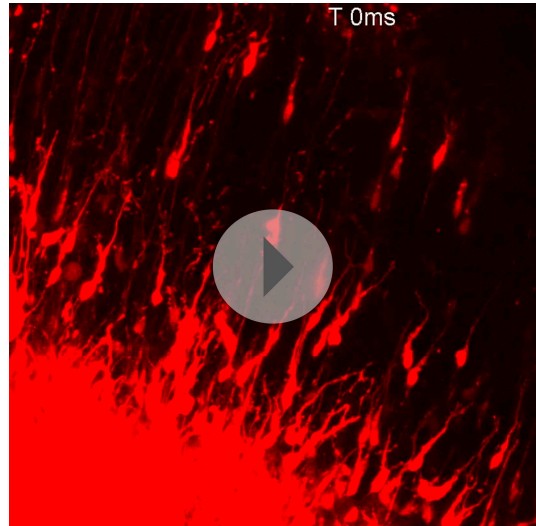

**Video 1**. Live cell imaging of radially migrating neurons in a cortical slice preparation. Control neurons electroporated at E15.5 migrate towards the pial surface after 3 days ex vivo. Neurons are imaged using time-lapse microscopy with images taken every 45 min for a 20-hr session.

cortical neuronal morphology, we deleted *Gsk3* using in utero electroporation of *Neurod1-Cre* as outlined above, and analyzed dendritic morphology at P15.

As demonstrated above, most *Gsk3*-deleted neurons failed to migrate and populated the deeper layers of the cortex at P15. Many of these migration arrested neurons exhibited abnormal dendritic arbors (data not shown). Because improper laminar position might affect dendritic arborization, we focused analysis on a small subset of *Gsk3*-deleted neurons that reached layer 2/3. Images at E16 (*Figure 1—figure supplement 1*) and E19 (*Figure 4A'*) show that a few *Gsk3*-deleted neurons are normally positioned and suggest that these normally positioned neurons did not undergo a substantial delay in migration. All of the neurons elaborated dendritic arbors and extended an axon into the corpus callosum. However, many of these normally positioned *Gsk3*-deleted neurons exhibited markedly abnormally oriented basal dendrites (*Figure 5A*, arrows). In many cases, basal dendrites were oriented towards the pial surface rather than towards the deeper cortical layers (*Figure 5B–D*). Additionally, *Gsk3*-deleted basal processes grew longer and were more branched than those of control neurons (*Figure 5D* and *Figure 5—figure supplement 1C,D*). Many *Gsk3*-deleted neurons also exhibited striking defects in the apical dendrite (*Figure 5C,C'*,E orange arrows and *Figure 5—figure supplement 1B*). Thus, apical dendrites of the *Gsk3*-deleted neurons, although properly oriented towards the pial surface often extended branches close to the soma that extended apically rather than laterally (*Figure 5E*, orange arrows and *Figure 5—figure supplement 1B*).

Perhaps surprisingly *Gsk3*-deleted neurons extended axons into the callosum towards the contralateral cortex (*Figure 4A,A'* orange arrows). Axonal arborization in the contralateral cortex had reduced density, an abnormality that is under further investigation (data not shown).

## GSK-3 regulation of migration is independent of Wnt/β-catenin signaling

Signaling via β-catenin is an obvious candidate to mediate GSK-3 regulation of migration. In the canonical Wnt cascade, Wnt signaling through frizzled receptors leads to dishevelled and GSK-3 sequestration, β-catenin accumulation and enhanced β-catenin/TCF-mediated transcription (*Kaidanovich-Beilin and Woodgett, 2011*). In radial progenitors, β-catenin signaling is clearly an important mediator of the effects of GSK-3 deletion on proliferation (*Chenn and Walsh, 2002*; *Kim et al., 2009*).

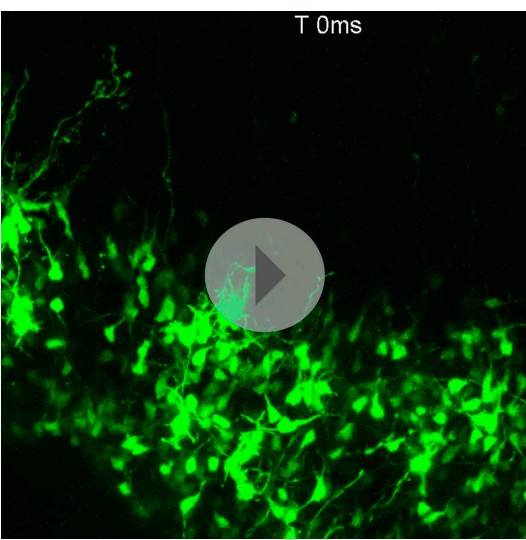

**Video 2**. Gsk3-deleted neurons fail to migrate in cortical slice preparation. *Gsk3a*<sup></sup>$Gsk3a^{-/-}Gsk3b^{loxp/loxp}$ mice electroporated at E15.5 with *Neurod1-cre* and Z/EG have an elongated multipolar stage and do not migrate towards the pial surface. Images were taken every 45 min over a 20-hr imaging session.

To determine the role of GSK-3 regulation of β-catenin in developing cortical excitatory neurons, we utilized a β-catenin (*Ctnnb1*) mouse that harbors loxP sites flanking exon 3 ($Ctnnb1^{Ex3}$: *Neurod6*) (*Harada et al., 1999*). Exon 3 encodes the residues that GSK-3 phosphorylates to signal β-catenin degradation; thus deleting exon 3 stabilizes β-catenin. Using a β-catenin antibody directed at the residues encoded by exon3 verified a reduction in this protein fragment at P0 as expected after *Neurod6*-mediated recombination (*Figure 6C,H*). Migration of Cux1 neurons was entirely normal in these animals (*Figure 6A*, *Figure 6—figure supplement 1A*). Further, in contrast to *Gsk3:Neurod6*, $Ctnnb1^{Ex3}$:*Neurod6* mice survive, breed, and have no overt behavioral phenotype. They display a rostral midline defect resulting from lack of the hippocampal commissure (data not shown), as seen in other models using stabilized β-catenin (*Chenn and Walsh, 2003*).

For additional assessment of the role of β-catenin signaling, we performed global analysis of GSK-3 transcriptional targets at E18 in control and *Gsk3:Neurod6* cortical lysates using Affymetrix microarray analysis. Consistent with loss of *Gsk3* in our conditional mutants, probe level information specific to exon 2 of *Gsk3b* showed that exon 2 was decreased by an average 2.15-fold. However, classic Wnt pathway target genes downstream of β-catenin/(TCF)/LEF-1 transcription factors, including CyclinD1, Brachyury, Wisp1, Cdx1, and Engrailed2 were unchanged (data made available in GEO, GSE58727). Finally mice in which β-catenin signaling is abrogated after *Neurod6-Cre* mediated recombination ($Ctnnb1^{loxp/loxp}$:*Neurod6*) also show no change in migration of Cux1 neurons (ES Anton, personal communication, June 2014).

To further assess a potential role of Wnt/β-catenin signaling in cortical lamination, we created a conditional dishevelled $2^{loxp/loxp}$ (*Dvl2*) mouse (*Ohata et al., 2014*). We then generated a triple mutant by crossing our floxed $Dvl2^{loxp/loxp}$ with existing *Dvl1* and *Dvl3* nulls and the *Neurod6-Cre* line (*Dvl123:Neurod6*). Deletion of all three *Dvls* presumably completely abrogates Wnt signaling via the canonical pathway. A robust decrease of DVL2 protein (91%) was observed in P0 cortical lysates from *Dvl123:Neurod6* mutants when compared to Cre<sup>−</sup> $Dvl2^{loxp/loxp}$ controls (*Figure 6C,H*). These mice are born but die immediately at P0. Remarkably, lamination in the triple allele mutant *Dvl123:Neurod6* appears relatively normal at E18 (*Figure 6B*). This result further supports the idea that GSK-3 regulation of migration is not mediated by the WNT/β-catenin cascade.

## Surprising lack regulation by STK11 (LKB1), CDC42, and PTEN

Recent work utilized RNAi and GSK-3 S9A mutant constructs to conclude that STK11 (LKB1) inactivation of GSK-3 via ser9 phosphorylation alters neuronal migration (*Asada and Sanada, 2010*). However, *Gsk3* knock-in S9A/S21A *Gsk3* 'constitutively active' mice develop normally and migration defects have not been reported (*Jiang et al., 2005*; *Gartner et al., 2006*). To further address the issue of GSK-3 inhibition downstream of STK11, we genetically deleted *Stk11* from developing excitatory neurons (*Stk11:Neurod6*). *Stk11:Neurod6* mutant mice die around P20. Western blot analysis verified a 66% decrease in STK11 (LKB1) protein in *Stk11:Neurod6* mutant mice when compared to heterozygous littermate controls (*Figure 6G,H*). Perhaps surprisingly, no gross lamination abnormalities were observed (*Figure 6D*, *Figure 6—figure supplement 1B*). Thus in our hands, STK11 (LKB1) regulation of GSK-3 activity is not essential for radial-guided neuronal migration in vivo.

We also deleted a key regulator of Par6-aPKC, *Cdc42*, using *Neurod6-Cre*. aPKCs are also known to phosphorylate Ser9/Ser21 of GSK-3 (*Etienne-Manneville and Hall, 2003*). *Cdc42:Neurod6* mutants also die shortly after birth, but again we observed no gross lamination defect in these mutant mice

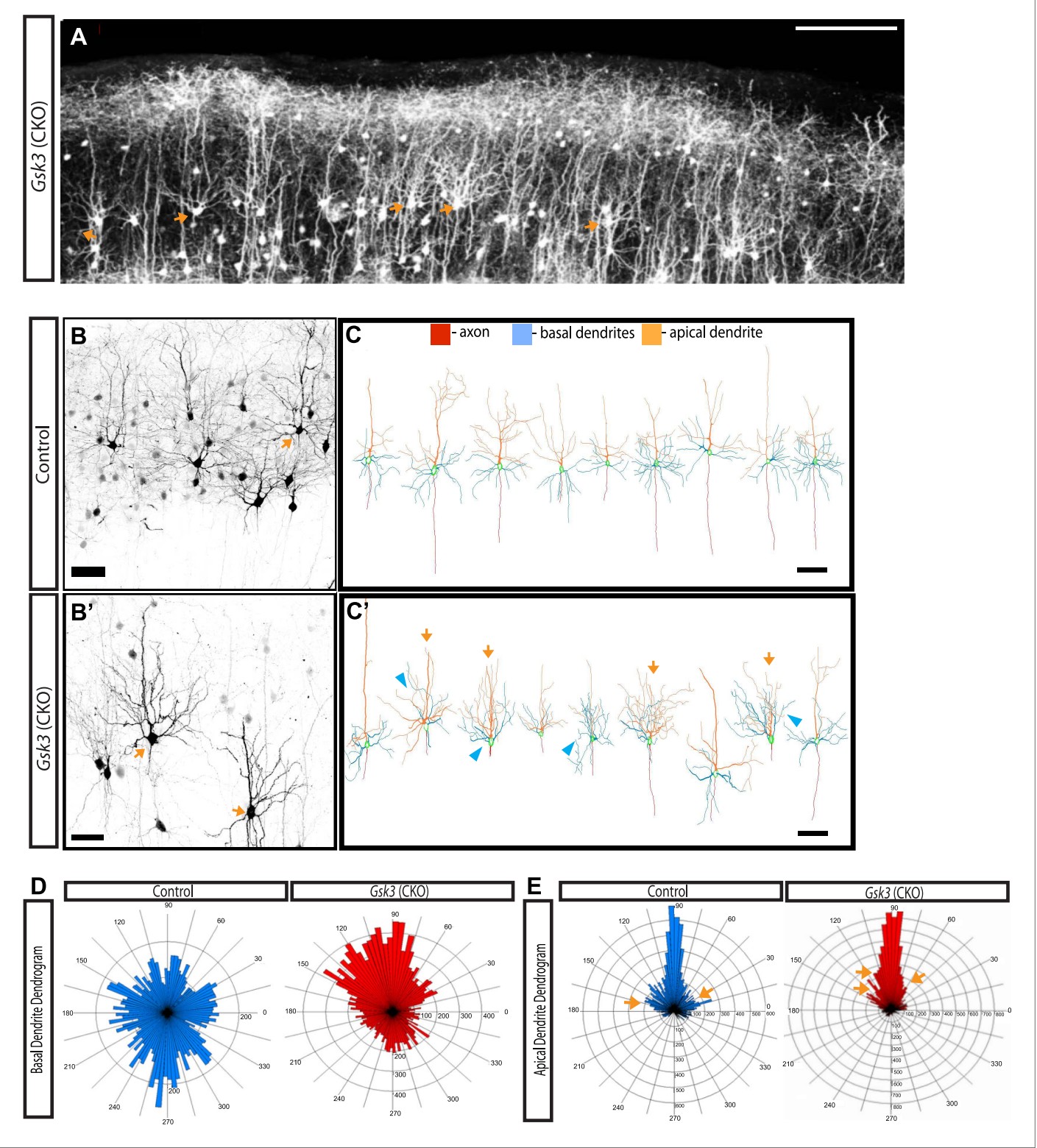

**Figure 5**. GSK-3 signaling is required for proper dendrite orientation. (**A**) Gsk3 deleted neurons at P15 shown after in utero electroporation at E14.5 with *Neurod1-Cre* and Z/EG plasmids. Multiple neurons with obvious abnormalities in dendritic orientation were observed in the upper layers of the cortex (orange arrows). Scale bar = 200 µm. (**B–B'**) Control and *Gsk3*-deleted neurons in the upper layers at P15, immunostained with antibodies against eGFP (black) using same methods as *Figure 4*. Gsk3-deleted neurons have abnormally polarized arbors indicated by orange arrows. Scale bar = 50 µm.
*Figure 5. Continued on next page*

*Figure 5. Continued*

(**C–C'**) Neurolucida reconstructions of control and *Gsk3*-deleted neurons in the upper layers of the cortex. The axon (red) projects towards the ventricle in control and *Gsk3*-deleted neurons. Both apical dendrites (orange) and basal dendrites (blue) are more branched (orange arrows) and basal dendrites (blue) are mispolarized (blue arrowheads) in *Gsk3*-deleted neurons. Scale bar = 100 µm. (**D**) Basal dendrite quantification. Dendrogram shows that basal dendrites more frequently project towards the pial surface in *Gsk3*-deleted neurons when compared to control basal dendrite orientation. (n = 3, n = 3 CKO; 15 control and 15 *Gsk3*-deleted neurons quantified). (**E**) Apical dendrite dendrogram indicates polarization and length of processes. Control apical dendrites project pially (90°). Numerous small apical branches form near the soma and project laterally (orange arrows). *Gsk3*-deleted neurons also project pially-directed apical dendrites. However, Apical branches have a pially-directed orientation, resulting in abnormal morphology (orange arrows, also see **C'**).

The following figure supplements are available for figure 5:

**Figure supplement 1**. Quantification of dendritic branching at P15 in control and *Gsk3-deleted* neurons.

(*Figure 6E*, *Figure 6—figure supplement 1C*). Western blot analysis verified an 87% decrease in CDC42 protein levels in mutants when compared to littermate wild type controls at P0 (*Figure 6G,H*).

In receptor tyrosine kinase (RTK) cascades, GSK-3 lies downstream of PI3K/ AKT signaling. Signals transduced through this cascade inhibit GSK-3 via phosphorylation on Ser9/Ser21 (*Hur and Zhou, 2010*). The phosphatase PTEN suppresses PI3K signaling. To determine if PI3K signaling affects migration, we genetically deleted *Pten* in neurons using *Neurod6-Cre*. Western blot analysis at P0 revealed an 88% decrease in PTEN protein in our conditional knockouts when compared to littermate wild-type controls (*Figure 6G,H*). Though *Pten:Neurod6* mutants die around birth, no overt lamination defects were observed (*Figure 6F*, *Figure 6—figure supplement 1D*).

These findings taken together suggest that regulation of GSK-3 phosphorylation at ser9/ser21 is not important in the control of radial migration.

## Reduced phosphorylation of DCX and CRMP-2 in GSK-3 mutants

To determine the status of relevant GSK-3 targets, we conducted western blot analysis of GSK-3 substrates that have been implicated in migration regulation. Recently, the key migration regulator doublecortin (DCX) was shown to be a GSK-3 substrate (*Bilimoria et al., 2010*). In P0 cortical lysates of *Gsk3^loxp^:Neurod6* mutants, we observed a 61% decrease in phosphorylated DCX on Ser327 (*Figure 7A–A'*). These results demonstrate that DCX is importantly regulated by GSK-3 in developing cortical neurons.

CRMP-2 family proteins have also recently been implicated in control of radial migration (*Ip et al., 2014*). We found an 80% decrease in phosphorylated CRMP-2 on Thr514 in P0 cortical lysates from *Gsk3^loxp^:Neurod6* mutants (*Figure 7A–A'*). Changes in the phosphorylation status of DCX and CRMP-2 have been implicated in migration control, raising the possibility that reduced functions of these proteins toward microtubules are responsible for the effects of *Gsk3* deletion.

GSK-3 regulation appeared to be surprisingly specific. Thus, we did not detect changes in pSer722 FAK (*Bianchi et al., 2005*), pSer744 dynamin-1 (*Clayton et al., 2010*), and pSer129 CREB (*Fiol et al., 1994*; *Bullock and Habener, 1998*) (*Figure 7B–B'*). Presumably other kinases contribute to phosphorylation of the putative GSK-3 sites in vivo. However, we cannot rule out changes that would be masked by the dilution effect of non-recombined cells, changes confined to specific cellular compartments, or changes that might be more apparent later in development. There was no increase in cleaved caspase-3 staining or other evidence of apoptosis in the mutants at P0 (*Figure 7B–B'*). We did see increases in cleaved caspase-3 staining in cortex in *Gsk3^loxp^:Neurod6* mice starting at later postnatal stages and these changes currently under investigation (data not shown).

## Discussion

### Overview

In this work, we have documented a cell autonomous requirement for GSK-3 signaling in migration and dendritic orientation of cortical excitatory neurons. GSK-3 activity is critical for the radial migration of later born, Cux1-expressing neurons in all regions of cortex, and for radial migration in the hippocampus. The GSK-3 requirement is specific for radial migration as tangential migration proceeded despite *Gsk3* deletion. GSK-3 regulation of migration appears to be independent of Wnt/β-catenin

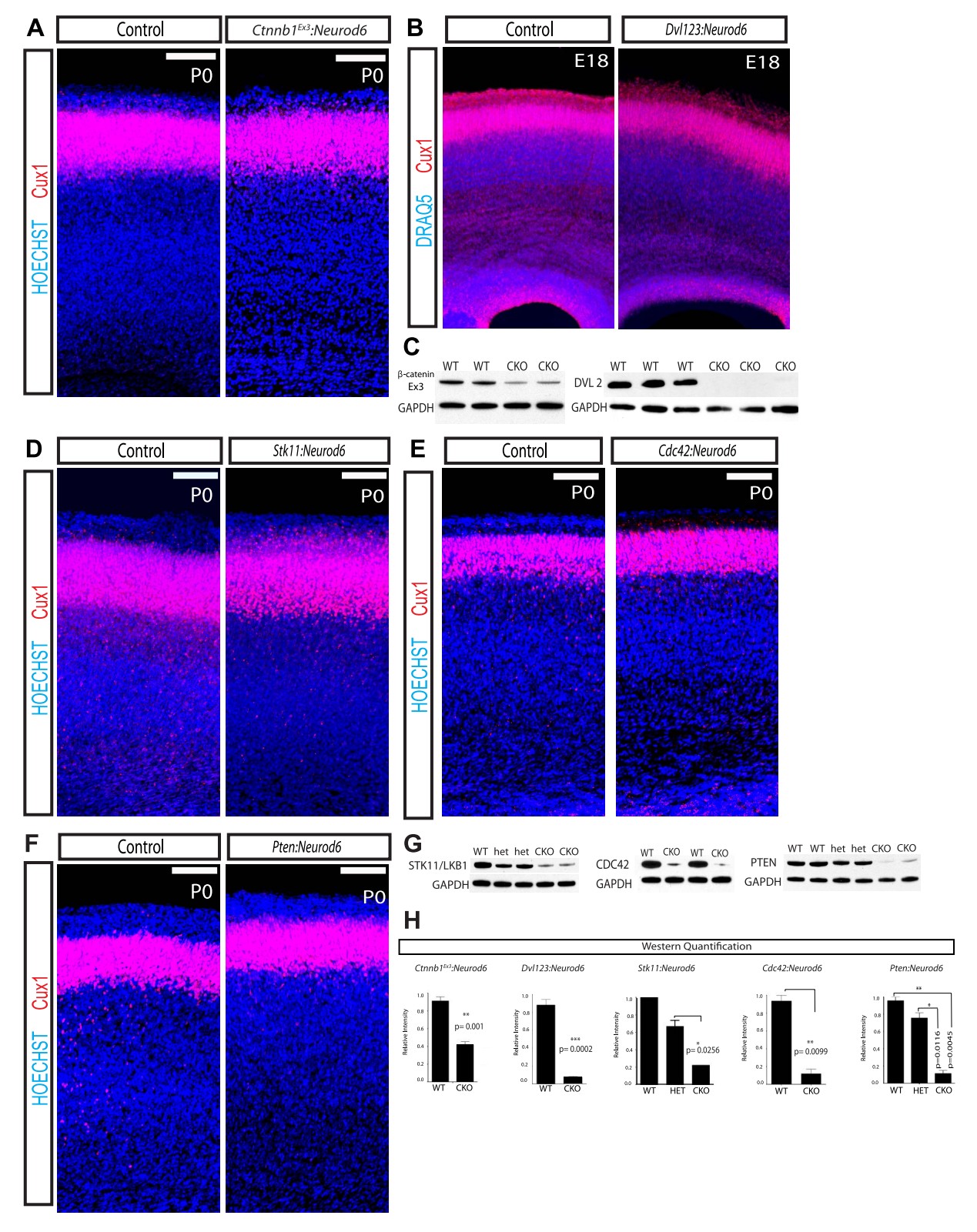

**Figure 6**. Lamination in other signaling mutants. (**A**) P0 representative coronal sections of control (heterozygous for floxed allele) and *Ctnnb1^{Ex3}:Neurod6* mutants stained for Cux1 (red) and Hoechst (blue). (n = 5) Scale bar = 100 μm. (**B**) E18 Coronal sections of control and *Dvl123:Neurod6* showing Cux-1 (red) and DRAQ5 (blue) staining. Cux-1 neurons reach layer 2/3 in both controls and *Dvl123:Neurod6* triple mutants. (n = 3). (**C**) Western blot verification

*Figure 6. Continued on next page*

*Figure 6. Continued*

of protein deletion after recombination of floxed alleles with *Neurod6-Cre*. *Ctnnb1^{Ex3}:NeuroD6* (n = 3 WT, n = 3 CKO), *Dvl123:NeuroD6* (n = 3 WT, n = 3 CKO). GAPDH was probed as a loading control. (**D–F**) P0 representative coronal sections of control (heterozygous for floxed allele) and indicated mutant lines stained for Cux1 and Hoechst. Scale bars are 100 μm, at least n = 5 per line. (**G**) Western blot verification of protein deletion in mutant lines. GAPDH was probed as a loading control. N's refer to numbers of mutants and paired controls. *Stk11:Neurod6* (n = 2), *Cdc42:Neurod6* (n = 2), *Pten:Neurod6* (n = 3). (**H**) P0 Western Blot quantification of *Ctnnb1^{Ex3}:Neurod6*, *Dvl123:Neurod6*, *Stk11:Neurod6*, *Pten:Neurod6* and *Cdc42:Neurod6* lines.

The following figure supplements are available for figure 6:

**Figure supplement 1**. Quantification of lamination in other signaling mutants.

and PI3K signaling. Rather, *Gsk3* deletion is associated with striking reductions in phosphorylation of key microtubule regulatory proteins, DCX and CRMP-2.

## GSK-3 regulation of cortical neuronal development

Our work builds on a growing body of evidence establishing critical requirements for GSK-3 in cortical neuronal development, with GSK-3 signaling having different functions at different developmental stages. In radial progenitors, GSK-3 is a critical mediator of proliferation via regulation of β-catenin (**Kim et al., 2009**). At later stages, recent work has demonstrated that GSK-3 is a key mediator of the

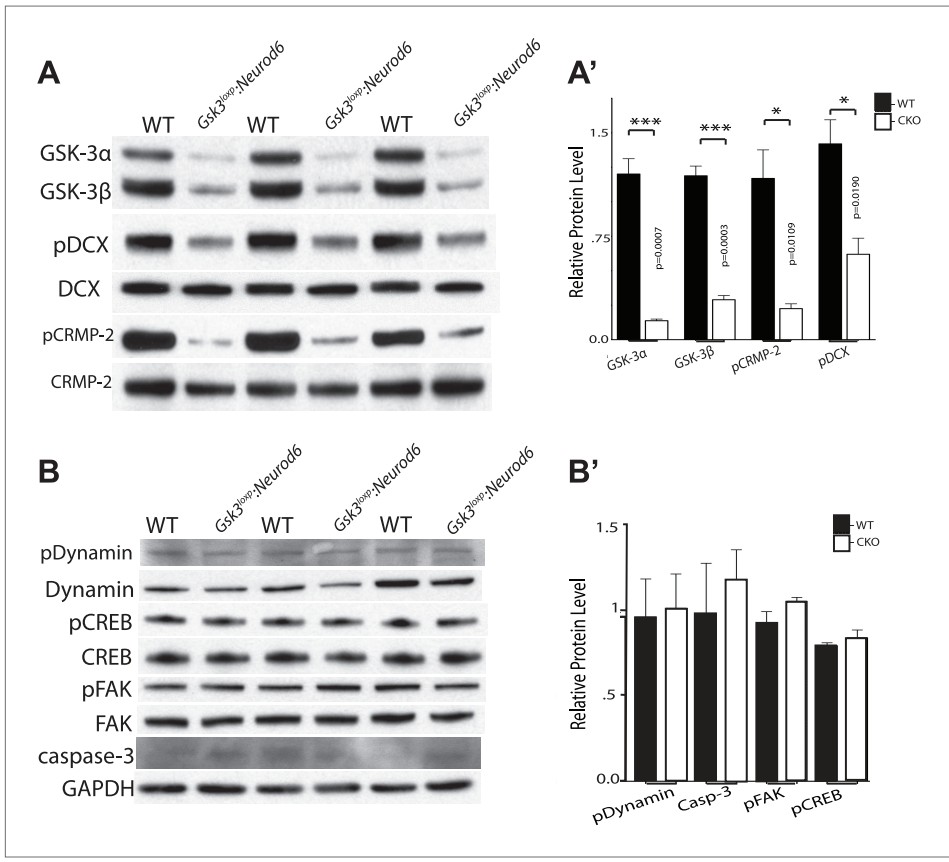

**Figure 7**. Phosphorylation status of GSK-3 substrates. (**A–A'**) Western blots of P0 cortical lysates from *Gsk3^{loxp}:Neurod6* mutants and wild-type controls performed in triplicate. Levels of GSK-3 proteins and phospho-target proteins are shown. Strong reductions in phosphorylation of doublecortin on ser327/Thr321 and CRMP-2 on Thr514 are evident. (**A'**) Quantification of relative densities from **A**. p values shown in figure (n = 3, unpaired t-test) (**B–B'**) Western blots of cortical lysates at P0 showing levels of other GSK-3 targets. No changes were observed in phosphorylation of dynamin, pCREB, or pFAK. No change was observed in cleaved caspase-3. GAPDH was used as a loading control. (**B'**) Quantification of relative protein densities (n = 3, unpaired t-test).

amplification of the INP pool via interactions with the scaffolding protein Axin (*Fang et al., 2013*). Axin-GSK-3 binding in the cytoplasm is clearly required for expansion of the INP pool, although the mechanism of GSK-3 action was not specified in this study. Interestingly, β-catenin regulation of transcription in the nucleus was required for differentiation of INPs into neurons. These effects of GSK-3 signaling on radial progenitors and INPs are presumably a strong determinant of final cortical neuronal number in mature animals.

We now find that excitatory neuron development is also under important GSK-3 regulation. Thus, Cux1-expressing neurons require GSK-3 signaling for timely multipolar to bipolar transition and migration along radial processes. Deeper layer neurons expressing Tbr1 were not strongly influenced, but we cannot be certain that GSK-3 protein was fully depleted in these earlier born neurons at the time migration was occurring. The behavioral consequences of migration failure and dendritic arbor abnormalities could not be assessed due to early death of the *Gsk3*-mutant animals. Whether death was due to cortical abnormalities or defects in other cells that underwent recombination with *Neurod6-Cre* could not be determined. However, even a very mild form of this type of migration defect would presumably be catastrophic for human brain development.

In contrast to regulation of progenitor proliferation and neural differentiation, GSK-3 regulation of radial migration appears to be independent of Wnt/β-catenin signaling. Thus, neither the stabilization of β-catenin nor deletion of all *Dvl*s in developing cortical neurons using the *Neurod6-Cre* driver produced major defects in cortical layering. Interestingly, the migration defect also appears to be independent from a migration defect associated with mutations of the schizophrenia-associated protein, DISC1. DISC1 regulates progenitor proliferation via interactions with GSK-3, decreased GSK-3 kinase activity and, ultimately, increased β-catenin signaling (*Mao et al., 2009*). Interestingly, a recent study demonstrated that DISC1 regulates migration independently of GSK-3 via effects on the centrosome (*Ishizuka et al., 2011*). However, our work reported here clearly demonstrates that GSK-3 also has a critical role in the regulation of cortical neuronal migration.

Radial migration is an enormously complex process requiring timely progenitor differentiation, dramatic morphological change, intricate mechanisms for cell and nuclear movements, and dynamic sensing of multiple cues that start and stop the process. Not surprisingly, GSK-3 joins dozens of other molecules implicated in the control of radial migration (see *Ayala et al., 2007* for review). Comparisons with the literature suggest that effects of GSK-3 deletion are among the most severe that have yet been observed.

Interestingly, GSK-3 importantly regulates multiple proteins implicated in migration and is an important mediator in several of the signaling pathways involved. Thus GSK-3 is known to phosphorylate DCX (*Bilimoria et al., 2010*), FAK (*Bianchi et al., 2005*), dynamin (*Clayton et al., 2010*), neurogenin (*Li et al., 2012*), CRMP-2 (*Uchida et al., 2005*; *Yoshimura et al., 2005*), and MAP1B (*Trivedi et al., 2005*). Further, GSK-3 mediates Reelin signaling (*Beffert et al., 2002*), LKB1 effects (*Asada and Sanada, 2010*), Cdc42 effects (*Etienne-Manneville and Hall, 2003*), integrin signaling (*Guo et al., 2007*), semaphorin signaling (*Eickholt et al., 2002*; *Uchida et al., 2005*), and other pathways that have been implicated in control of radial migration. Finally, GSK-3 shares multiple substrates with cdk5, a kinase that is situated among the most important regulators of neuronal migration and acts as a 'priming' kinase for GSK-3 signaling (*Xie et al., 2006*).

Clearly GSK-3 effects on neuronal development extend well beyond the phase of migration. For example, although dendritic arbors form, abnormalities in dendritic orientation were striking in *Gsk3*-deleted neurons. Importantly, interference with semaphorin signaling mediators also produces abnormalities of apical process development and orientation of basal dendrites (see below).

## Mechanisms of GSK-3 regulation

In general, GSK-3 acts via two classes of mechanisms: one where GSK-3 activity inhibits substrate function or availability and another where GSK-3 activity is required for substrate function. Therefore, we might expect *Gsk3* deletion to enhance processes normally inhibited by GSK-3 activity and to inhibit processes that require GSK-3 activity.

Multiple studies have suggested that inhibition of GSK-3 kinase activity is important for morphological functions such as establishment of neuronal polarity and cellular migration. The effects of GSK-3 inhibition are thought to be mediated by dephosphorylation of CRMP-2, APC and other cytoskeletal mediators with resulting stabilization of microtubules at the tips of axons (*Etienne-Manneville and Hall, 2003*; *Jiang et al., 2005*; *Yoshimura et al., 2005*; *Hur and Zhou, 2010*). Another recent study

employing in utero electroporation suggested that an STK11(LKB1)/GSK-3 pathway resulting in GSK-3β ser9 phosphorylation and APC localization at the leading edge was important in cortical neuronal migration (**Asada and Sanada, 2010**).

In the studies outlined above, inhibition of GSK-3β kinase activity is indicated by phosphorylation on Ser9 and inhibition of GSK-3α by phosphorylation on Ser21. However, a major unresolved paradox is that mice *with Gsk3a and Gsk3b* point mutation knock-ins that prevent ser9 and ser21 phosphorylation respectively do not exhibit widespread morphological abnormalities (**Gartner et al., 2006**). Similarly, truncation mutants of the *Drosophila* GSK-3 homologue, *Shaggy*, that lack inhibitory phosphorylation sites are not associated with morphological abnormalities (**Papadopoulou et al., 2004**). In the context of neuronal migration, although GSK-3 ser9/21 phosphorylation is enhanced downstream of Reelin signaling (**Beffert et al., 2002**; **Gonzalez-Billault et al., 2005**), GSK-3β kinase activity towards the important substrate MAP1B is actually increased rather than diminished (**Gonzalez-Billault et al., 2005**). Finally, our findings reported here do not support critical roles in radial migration for STK11, CDC42, or PTEN, all of which are known to regulate GSK-3 ser9/21 phosphorylation. These findings, taken together, raise questions about the importance of negative regulation of GSK-3 via Ser9/21 phosphorylation on functions of microtubule binding proteins related to migration in vivo.

Our results demonstrate that *Gsk3* deletion blocks radial migration and that *Gsk3* over-expression may enhance it. Thus our results are more in line with a mechanism in which GSK-3 activity is required for normal substrate function. Several proteins have been shown to require GSK-3 kinase activity for normal function. These include DCX where GSK-3 mediated phosphorylation is thought to be required for DCX's actions in regulating axon branching and possibly in migration (**Bilimoria et al., 2010**). Additional examples possibly relevant to migration are GSK-3 regulation of FAK phosphorylation at ser722 (**Bianchi et al., 2005**) and dynamin-1 phosphorylation of ser774 (**Clayton et al., 2010**). GSK-3 phosphorylation regulates functions of these proteins in complex ways, but possibly enhances functions that may be required for neuronal migration.

A well-studied example of a pathway that is associated with upregulation of GSK-3 activity is semaphorin signaling. Semaphorin family members have multiple roles as chemorepellants and chemoattractants for many classes of axons and dendrites (for review see **Pasterkamp, 2012**). Semaphorin signaling strongly activates GSK-3 (**Eickholt et al., 2002**) resulting in phosphorylation of a key MAP, CRMP-2 (**Uchida et al., 2005**), as well as other CRMP family members (**Cole et al., 2006**; **Yamashita and Goshima, 2012**). Intriguingly, during neuronal polarization semaphorin signaling is associated with suppression of axons, and formation of dendrites (**Shelly et al., 2011**), consistent with the idea that dephosphorylated CRMP-2 may favor axon formation, whereas phosphorylated CRMP-2 may be critical to proper formation of dendritic arbors.

Importantly, migration and dendritic abnormalities that we have described here are consistent with cortical neuron phenotypes reported due to manipulation of semaphorin and CRMP signaling. Interfering with semaphorin, NP1 or Plexin via silencing RNAs and gene knockouts has been reported to interfere with radial migration (**Chen et al., 2008**; **Renaud et al., 2008**). Application of exogenous semaphorin supports apical process formation (**Polleux et al., 2000**). Further, reduced semaphorin signaling causes bifurcation of CA1 pyramidal dendrites in the developing hippocampus (**Nakamura et al., 2009**). Downstream of semaphorins, interfering with CRMP functions has been reported to affect migration and result in branched apical dendrites and abnormal dendrite orientation (**Yamashita et al., 2012**; **Yamashita and Goshima, 2012**; **Ip et al., 2012**; for a review see, **Ip et al., 2014**). Finally, in the context of cortical neuronal migration, phosphorylation of CRMP-2 promotes migration. Thus a GSK-3 phosphomimetic CRMP-2 can rescue migration defects associated with knockdown of the semaphorin signaling mediator, α2-chimaerin (**Ip et al., 2012**). Taken together, these findings suggest that inhibition of CRMP2 phosphorylation at Thr514 in *Gsk3*-deleted neurons may partially explain the migration and dendritic orientation abnormalities we have observed.

## Conclusion

In sum, we have demonstrated a key cell autonomous role for GSK-3 signaling in regulating radial migration and dendritic orientation of cortical excitatory neurons. *Our Gsk3* deletion results are not in line with what might have been predicted from prior studies that have correlated ser9/21 phosphorylation with relief of negative GSK-3 phosphorylation and negative regulation of cytoskeletal-associated proteins. In particular, we do not find evidence that an STK11(LKB1)/GSK-3 inhibitory pathway is a key migration regulator. In contrast, our results are consistent with the idea GSK-3 phosphorylation of DCX

and CRMP-2 is required for appropriate cytoskeletal regulation and migration. Finally, our data are consistent with the idea that GSK-3 activity is essential for the effects of semaphorin and CRMP-2 signaling on cortical neuronal development in vivo.

## Materials and methods

### Mice

Mice were cared for according to animal protocols approved by the Institutional Animal Care and Use Committees of the University of North Carolina at Chapel Hill. GSK-3 lines were generously provided by Jim Woodgett. *Gsk3a* mice possessing exon 2 deletions and *Gsk3a* and *Gsk3b* loxp flanked exon 2 mice have been previously described (*Doble et al., 2007*; *MacAulay et al., 2007*; *Patel et al., 2008*). *Gsk3:Neurod6* mice were generated by mating *Gsk3a$^{-/-}$*, *Gsk3b$^{loxp/loxp}$* with *Neurod6:Cre* mice generously provided by Dr KA Nave (*Goebbels et al., 2006*). *Gsk3* mutant mice and *Neurod6-Cre* mice were maintained on mixed genetic backgrounds. For all experiments, four allele mutants were compared with littermate controls. Triple allelic male mutants (*Gsk3a$^{+/-}$b$^{f/f}$:Neurod6*) fail to survive into adulthood with death occurring around P25 for reasons not yet determined.

*Ctnnb1* exon 3 (*Ctnnb1$^{Ex3}$*) floxed mice were previously described (*Harada et al., 1999*) and maintained on a mixed C57/Bl6 129 background. *Stk11*-floxed mice have been previously described (*Bardeesy et al., 2002*) and were maintained on a mixed C57/Bl6 129, CF1 background. *Pten*-floxed mice (*Groszer et al., 2001*), *Cdc42*-floxed mice were purchased from Jackson lab and have been previously described (*Chen et al., 2006*).

*Dvl2* floxed allele was generated in the UNC Neuroscience Center Molecular Neuroscience Core using conventional methodology (*Ohata et al., 2014*). *Dvl1* (*Beier et al., 1992*) and *Dvl3* (*Tsang et al., 1996*) mutant mice were purchased from The Jackson Laboratory (Bar Harbor, Maine) have been previously described. All *Dvl* mutant mice were maintained on a mixed background.

Results from mutants and control littermates shown in all of the figure panels were based on at least three experiments from independent litters, unless otherwise noted.

### In utero electroporation

Mice were anesthetized using 2,2,2-Tribromoethanol (4 mg/10 g mouse) or isofluorine, and embryos were exposed at E14.5 and E15.5. Plasmids were mixed with fast green and microinjected into the lateral ventricle of embryos using a picospritzer. Embryos were electroporated with five 50 ms pulses at 30–35V with a 950 ms interval and returned to the abdominal cavity. Plasmids used were: *Neurod1-Cre*, *lacZ/Egfp*, lox-STOP-lox Ai9, *Neurod1-Gsk3b*, and *Neurod1-EGFP*. A CF1 foster dam was used to aid postnatal survival studies. Depending on the experiment, mice were perfused at E19.5, P0, P10, or P15 with 4% paraformaldehyde (Sigma-Aldrich, St. Louis, MO) in phosphate buffered saline (PBS) and post fixed overnight.

### Ex vivo electroporation and organotypic cortical slice culture

Cortical progenitor cells were electroporated ex vivo at E15.5 as described previously (*Hand et al., 2005*). Briefly, E15.5 embryos were decapitated, plasmids were injected into lateral ventricle followed by electroporation with four 30V pulses that were 30–40 ms in duration and separated by a 100-ms interval. Following electroporation, brains were dissected and vibratome sectioned at 250 μm. Slices were transferred to Poly-D-Lysine and laminin coated culture insert (Millipore, Billerica, MA) in a FluoroDish (World Precision Instruments, Sarasota, FL), then 2 ml Basal Medium Eagle with FBS, N2 (Gemini Bio-Poducts, West Sacramento, CA), B27 (Life Technologies, Grand Island, NY), penicillin-streptomycin (Life Technologies), and L-glutamine (Life Technologies) supplements were added. Slices were cultured for 3 days at 37°C and live imaged using an Olympus FV1000 Confocal microscope with stage incubator.

### Cortical progenitor cultures

E14.5–15.5 dorsal cortices were electroporated with *Neurod1-Cre*; Z/EG, dissected in 4°C Hank's Balanced Salt Solution (HBSS) (Life Technologies) and dissociated into single cells using Trypsin (Life Technologies) according to previously described methods (*Hand et al., 2005*). Neurons were plated on glass bottom dishes (MatTek) coated with 0.1 mg/ml Poly-D-Lysine (Sigma) and 5 μg/ml Laminin (Sigma). Cells were cultured in Neuralbasal-A Medium (Life Technologies), supplemented with 1X B-27 (Gibco, 17,054-044), L-glutamine (Life Technologies), penicillin-streptomycin (Life Technologies), N2 (Life Technologies), and FBS. Neurons were fixed with 4% PFA and stained for stage progression analysis.

## Western Blotting

Mouse cortices were dissected from three control ($Gsk3a^{-/-}Gsk3b^{loxp/+}$:$Neurod6$) and mutant ($Gsk3$:$NeuroD6$) or three control ($Gsk3a^{loxp/loxp}Gsk3b^{loxp/loxp}$) and three mutant ($Gsk3a^{loxp/loxp}Gsk3b^{loxp/loxp}$:$NeuroD6$) mice from independent litters, collected in RIPA lysis buffer supplemented with protease and phosphatase inhibitors and cleared by centrifugation. Proteins were separated on SDS-PAGE gradient gels, transferred to a PVDF membrane and probed for proteins of interests and secondary HRP-conjugated antibodies were used for detection. Blots were washed and detection was performed with a commercially available ECL kit. ImageJ software (NIH) was used for quantification of band intensities, and intensities were normalized with total protein or the load control. Statistical analyses were conductive using Prism (GraphPad Software Inc., La Jolla, CA). Differences were considered statistically significant at $p<0.05$ and are included in each figure. The following antibodies were used: GSK-3 (Invitrogen, Carlsbad, CA), Actin (Cell Signaling Technology Inc., Danvers, MA), pFAK ser722 (Santa Cruz Biotechnology, Inc., Dallas, TX), FAK (Cell Signaling Technology Inc), pCRMP-2 Thr514 (Cell Signaling Technology Inc), CRMP-2 (Cell Signaling Technology Inc), pCREB ser129 (Santa Cruz Biotechnology, Inc.), CREB (Santa Cruz Biotechnology, Inc.), pDCX ser327, DCX (Cell Signaling Technology Inc.), pDynamin-1 ser744 (Santa Cruz Biotechnology, Inc.), Dynamin-1 (Santa Cruz Biotechnology, Inc.), PTEN (Cell Signaling Technology Inc.), LKB1 (Upstate), β-catenin (Cell Signaling Technology Inc), DVL2 (Cell Signaling Technology Inc), GAPDH (Cell Signaling Technology Inc), Cleaved Caspase-3 (Cell Signaling Technology Inc).

## Immunohistochemistry

Briefly, 100–150 µm free-floating vibratome sections of brains were collected in PBS, blocked with 5% normal serum in PBS with 0.1% Triton X-100 and incubated with primary antibodies in blocking solution overnight at 4°C. Cryosectioned tissue samples were immunostained as previously described (see *Newbern et al., 2011*). P15 in utero vibratome sections were blocked with 5% normal serum in PBS with 0/1% Triton X-100 and 2% DMSO, and incubated with primary antibodies for 3 days. Slices were rinsed with PBST for 24 hr and incubated with secondary antibodies for 3 days in PBS with 0/1% Triton X-100.

The following primary antibodies were used in our study: L1 (Millipore), Cux1 (Santa Cruz Biotechnology, Inc.), Neuronal Nuclei (Millipore), GFP (Aves Labs, Inc., Tigard, OR), CTIP2 (Abcam, Cambridge, MA) DRAQ5 (Fisher), TBR1 (Abcam, Cambridge, MA), HOECHST (Sigma), RFP (Rockland, Gilbertscille, PA). After rinsing, sections were then incubated with Alexa-conjugated secondary antibodies (Invitrogen) overnight at 4°C, rinsed three times in PBS, and mounted with gel/mount (Sigma).

## Quantification of cortical layering

*Gsk3* overexpression analysis was conducted with *Neurod1-Cre* and lox-STOP-lox-Ai9 plasmids as control and *Neurod1-Gsk3b* and *Neurod1*-eGFP plasmids for overexpression. Electroporations were performed at E14.5 and analyzed at E19.5. For controls and *Gsk3* over-expression, multiple comparable cortical sections were analyzed. For analysis of layering, the cortical plate was equally divided into eight bins, and cells in each bin were counted using ImageJ software. Bin 7–8 included layers 1–3. The percentage of GFP+ neurons in each bin were determined from multiple comparable sections.

Postnatal day 10 analysis after in utero electroporation was conducted with $Gsk3a^{+/-}Gsk3b^{loxp/loxp}$ control $Gsk3^{-/-}Gsk3b^{loxp/loxp}$ embryos. EGFP-filled neurons were counted and localized to either upper or deeper layers of cortex. Differences were considered statistically significant at $p \leq 0.05$ using an unpaired t-test.

Lamination quantification of mutant mice at P0 ($Gsk3^{loxp}$:$Neurod6$, $Gsk3$:$Dlx5/6$, $Ctnnb1^{Ex3}$:$Neurod6$, $Dvl123$:$Neurod6$, $Stk11$:$Neurod6$, $Cdc42$:$Neurod6$, $Pten$:$Neurod6$) was conducted using heterozygous controls (Cre+, loxP/+) compared to conditional knockouts. Vibratome sections (3–5 per mouse) were imaged at 40X with a 2.4-µm optical slice and Cux-1 cells were counted in an 8 bin analysis. Data were quantified as average percentage of Cux-1 neurons per bin with differences considered statistically significant at $p \leq 0.05$ using an unpaired t-test. Note that the histograms for controls *Figure 2* and *Figure 6—figure supplement 1* (based on a total of 10 control animals) all look very similar reflecting the fact that almost all Cux1 neurons have reached Layer 2–3 by P0 in control mice. The histogram for the *Gsk3* mutants in *Figure 2* is drastically different and reflects the distribution of Cux 1 neurons observed by inspection in many additional *Gsk3*:*Neurod6* and *Gsk3^{loxp}*:*Neurod6* mice.

In contrast, the histograms showing Cux1 neuron distribution for the other mutants were indistinguishable from controls.

Determination of number of Cux1 neurons at P0 in $Gsk3^{loxp}$:$Neurod6$ mice with $Gsk3a^{loxp/loxp}Gsk3b^{loxp/WT}$:$Neurod6$ heterozygous as controls was performed as follows: Three 100-μm thick vibratome sections were imaged and one to two 100-μm cortical columns were obtained per slice for a total of five columns per mouse. Total Cux1 cells and total cell numbers were counted using an 8 bin analysis with ImageJ software. Data were quantified as average percentage of Cux1/total per bin and differences were analyzed using an unpaired t-test.

## Image Acquisition and analysis

Images were collected on Zeiss LSM 710 and Zeiss LSM 780 confocal microscopes. Z-stack images were collected with 10X or 20X objectives and tiled together to generate high-resolution images of whole brain sections. Binned quantification images were taken using a 20× objective and a 2-μm optical slice. Randomly identified sections of the electroporated area in presumptive somatosensory cortex were imaged for quantification. Live imaging of migrating neurons was acquired on an Olympus FV1000 Confocal microscope with stage incubator. On average, 40 μm Z-stack images were acquired and merged every 30–60 min for 8–20 hr. A 40X objective at a 0.8-μm optical slice was used for GSK-$3^{loxp}$:Nex cux-1 cell counting.

## Dendritic arborization

P15 in utero experiments used $Gsk3a^{+/+}Gsk3b^{loxp/loxp}$ control $Gsk3a^{-/-}$ $Gsk3b^{loxp/loxp}$ or $Gsk3a^{loxp/loxp}$ $Gsk3b^{loxp/loxp}$ embryos. Images for the dendritic arbor reconstruction were acquired on a Zeiss LSM 7 MP multiphoton system using a W-PlanApochromat 20x/1.0 IR-corrected water immersion objective. On average, Z-stacks were acquired from 90 to 100 μm range with 0.65 μm steps. Dendritic reconstructions were performed using Neurolucida. A total of 15 control and 15 $Gsk3$-deleted cells from three mice each were reconstructed for dendritic analysis. Apical and basal dendrite polarity was quantified by generating dendrograms (Neurolucida) from reconstructed images. Sholl analysis calculated the number of dendrite intersections and dendritic lengths, Sholl quantification was conducted with an initial 15-μm somal radius and 20 μm concentric radial steps.

## Microarray

Dorsal cortices were dissected from three E18 control ($Gsk3a^{-/-}Gsk3b^{loxp/loxp}$) and mutant ($Gsk3;Neurod6$) embryos derived from two independent litters. RNA was prepared using the MiRNAeasy kit (Qiagen, Valencia, CA) and analyzed using the Affymetrix Mouse Gene 2.0 St array (Affymetrix, Santa Clara, CA). Following scanning of the array, basic data analysis was carried out using the Partek Genomics Suite Version 6.12.0712 (Partek, Inc., St. Louis, MO) Transcripts up or down-regulated by 1.5-fold were considered interesting candidates. Microarray data have been made available through Gene Expression Omnibus, accession number GSE58727 (GEO, NCBI).

## Acknowledgements

The authors would like to thank all members of the Snider Lab for thoughtful discussion. Dr Franck Polleux (Columbia University) provided the $Neurod1$-$Cre$, Lox-STOP-lox $lacZ/Egfp$ (Z/EG) plasmids, and $Stk11$ mutant mice. The phospho-DCX ser327/Thr321 antibody was provided by Dr Orley Reiner (Weizmann Institute of Science, Israel) and has been previously described (*Gdalyahu et al., 2004*).

## Additional information

### Funding

| Funder | Grant reference number | Author |
| --- | --- | --- |
| National Institutes of Health | NS050968 | William D Snider |
| National Institutes of Health | 5F31NS067838 | Meghan Morgan-Smith |
| National Institutes of Health | NS045892 | William D Snider |

The funders had no role in study design, data collection and interpretation, or the decision to submit the work for publication.

## Author contributions
MM-S, Conception and design, Acquisition of data, Analysis and interpretation of data, Drafting or revising the article, Contributed unpublished essential data or reagents; YW, Acquisition of data, Drafting or revising the article, Contributed unpublished essential data or reagents; XZ, Conception and design, Acquisition of data; JP, Acquisition of data, Contributed unpublished essential data or reagents; WDS, Conception and design, Analysis and interpretation of data, Drafting or revising the article

## Author ORCIDs
Meghan Morgan-Smith, http://orcid.org/0000-0001-5122-2330

## Ethics
Animal experimentation: This study was performed in strict accordance to animal protocol (#14-006) approved by the Institutional Animal Care and Use Committees (IACUC) of the University of North Carolina at Chapel Hill.

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
