## [Decision Letter]

Thank you for choosing to send your work entitled “GSK-3 signaling in developing cortical neurons regulates migration and dendritic polarization” for consideration at *eLife*. Your full submission has been evaluated by a Senior editor and 3 peer reviewers, one of whom is a member of our Board of Reviewing Editors, and the decision was reached after discussions between the reviewers.

While all three of the reviewers appreciated the general high quality of the data that were presented in this manuscript, after a lengthy discussion our concern was that, as is, the data presented in the manuscript are not sufficiently novel for publication in *eLife.* However, given the high quality of the manuscript, and potential interest of the findings, we would be happy to consider a revised version of the paper in which you have added perhaps another piece of data that adds a conceptual advance as well as data addressing the reviewers concerns. In that regard, if you were to validate all of the different conditional mouse knockout models that you have used with specific regard to their efficacy, then all of the reviewers agreed that this would considerably increase the novelty of the conclusions that you could draw.

*Reviewer #1:* In this manuscript by Morgan-Smith et al., the authors have defined an important role for GSK-3 signaling in the migration of excitatory cortical and hippocampal neurons during embryonic murine development. In general, the studies are very nicely performed, and the amount of work is extensive. Having said that, some of their conclusions do not seem to be justified on the basis of the data that are presented, and the manuscript could be significantly improved by consideration of a number of issues, as follows.

1) The authors demonstrate a robust and clearly-defined deficit in the migration of newborn Cux1-positive cortical neurons as soon as they are born. In contrast, they see that the deeper-layer Tbr1-positive neurons are unaffected. Since these neurons are born quite a bit earlier, are the authors confident that their driver line Nex-Cre actually causes recombination early enough to affect these earlier-born neurons? Also, most Tbr1-positive neurons are generated directly by radial glia, while the Cux1-positive neurons are largely generated via intermediate progenitors. Since intermediate progenitors are targeted by their Nex-Cre construct, might this provide an explanation for these differences?

While it is not essential for the authors to address these issues experimentally here, they should be more cautious in their interpretation of these differences.

2) In Figure 2, using mice floxed for both GSK3A and B, it looks like there are many more Cux1-positive neurons in addition to their deficit in migration. Is it possible that inducible deletion of GSK3 in intermediate progenitors causes an expansion of cortical neurons which in turn affects their migration indirectly? The authors need to address this possibility by quantifying the relative proportions of Cux1-positive neurons in these floxed mice relative to the controls.

3) In Figure 2, the authors show statistical significance, but they only have an n = 2 for controls. How can this be the case? I realize that some authors count each section as an n, but this isn't really valid for these types of analyses.

4) One of the main conclusions of this manuscript is that GSK3 is essential for radial but not tangential migration, based upon their analysis of cortical excitatory vs MGE-generated inhibitory interneurons. This conclusion isn't really warranted by the data they show. In particular, while Figure 3 does show that inhibitory interneurons make it into the cortex, these data need to be quantified and it needs to be shown that GSK3 has actually been deleted in these neurons to draw this conclusion.

5) Similarly, the authors conclusions about direct effects of GSK3 ablation on dendritic orientation are also difficult to draw based upon their experimental design. In particular, while I agree that their data convincingly show inappropriate orientation of the dendrites of Cux1-positive neurons in layers 2-3, the authors cannot rule out the possibility that these neurons were delayed or perturbed in their migration to this location, and that the dendritic deficits are a secondary consequence of any such perturbations. The authors need to deemphasize their conclusions that these are direct effects unless they have data dissociating migration from later events such as dendritic development.

6) Data in Figures 6 and 7 showing negative results vis a vis migration with multiple signaling proteins are a nice addition to the manuscript. However, to really substantiate these negative data, it would be important to (i) show that the Cre-mediated recombination really did occur as predicted, and (ii) show some kind of quantification to rule out the possibility that there are indeed neuronal deficits, but that they are just more subtle than with GSK3 knockout.

7) I'm not sure the small amount of microarray data shown in Figure 6 really adds anything to the manuscript. I would either add all of the microarray data or remove this.

8) The data on Crmp-2 are important and interesting. The Dcx data are potentially also very interesting, but the authors don't show that there is no change in total Dcx levels. In the absence of such data, it is difficult to know whether relative phosphorylation of Dcx is actually decreased.

*Reviewer #2:* In the manuscript 26-02-2014-RA-eLife-02663 entitled “GSK-3 signaling in developing cortical neurons regulates migration and dendritic polarization”, Morgan-Smith and colleagues address the question of molecular mechanisms of neuronal migration, a timely and very interesting problem with many significant knowledge gaps. Since many of the molecules implicated in this process are major players in important intracellular signalling events, this question is of considerable interest to a general audience. Specifically, the manuscript focuses on the regulation of cytoskeleton dynamics in migrating posmitotic neurons in the vertebrate cortex, and the role of LKB1, GSK-3 and APC signalling cascade. These proteins have been previously implicated in neuronal migration in a study using non-genetic and acute manipulations (Asada, N. & Sanada, 2010).

Taking advantage of the Nex:cre driver, the authors first analyse two different post-mitiotic neuron knockouts of GSK-3 alpha and beta and show that they are required for the positioning of Cux1+ neurons in cortical layers 2/3. Overexpression of GSK-3 through in utero electroporation results in more neurons occupying superficial cortical layers. Tangential neuronal migration in GSK-3 mutants is not affected suggesting that it GSK-3 is specifically involved in radial migration. Cell-autonomous deletion of GSK-3 reproduced the migration defect and revealed a neuronal morphology defect (the authors argue for dendrite orientation defects), as well as the persistence of the migration defects in postnatal animals. Next, the authors used Nex:cre to delete beta catenin, disheveled, LKB1, CDC42 and Pten, and show no effect on cortical neuron migration. However, the authors note, that in such knockouts, there was no change of Wnt target gene expression, or in the phosphorylation state of some of the GSK-3 substrates. However, the authors do finally show that GSK-3 loss of function does lead to a consistent decrease of pDCX and pCRMP-2 levels.

The data are mostly well-documented and convincing. Indeed, as authors claim, to my knowledge, this is the first genetic demonstration that GSK-3's are required for specific aspects of neuronal migration. My main concern is with the novelty of the findings, relative to the J Nsci study by Asada and Sanada which also showed a requirement for GSK-3 in migration, albeit through an acute, non-genetic manipulation. Other studies, including Shi et al. Curr Biol 14, 2025-2032 (2004) have also implicated GSK-3 in dendrite vs axon polarity.

Additionally, while I appreciate the lack of migration phenotypes in other KOs examined here, contrasted with Asada's manipulations of LKB1, I am concerned about the extent of the loss of function induced. The fact that in the b-cat KO, the authors fail to detect any effect on Wnt signalling targets makes me wary of their strong conclusions. Especially, since the effect of KO on b-catenin expression is barely more than a two-fold change. How do we know that the induction of those KOs is efficient enough to warrant the strong conclusion that LKB1, Pten, b-catenin etc, are not required for migration?

*Reviewer #3:* The study by Morgan-Smith and colleagues investigates the role of GSK3 in the developing mouse telencephalon. Using a line allowing efficient deletion of GSK3 in newly-born excitatory neurons, the study documents migration failures both in the developing cerebral cortex and the hippocampus. These defects are not observed with medial eminence-derived interneurons. The conclusion that GSK play a critical role in the migration of cortical excitatory neurons is supported by data based on a GSK-overexpression approach causing enhanced migration. A thorough analysis of the canonical Wnt pathway suggests that it is not involved in the phenotype described here. By contrast, based on results illustrating a dramatic reduction of phosphorylated CRMP-2 it would appear that GSK may be critically involved in a semaphorin pathway. These results fit well with recent observations by Ip and colleagues as well as with a report by Chen and colleague on the role of Semaphorin 3A in the radial migration of cortical neurons. This is a very well executed, thorough and interesting study. In addition to the main results summarised in the above, another virtue of this study is that it reports on a range of negative, but surprising and useful results. Beyond the lack of involvement of the Wnt pathway, the lack of effects of the deletions of PTEN and especially of Cdc42 are quite surprising. Clearly then, this important study should be published in a scholarly, high profile journal as it is likely to be of high interest to developmental neurobiologists as well as to human neurogeneticists.

My main recommendation would be to revise the Abstract to better highlight the interest of the study. In particular, the first sentence is difficult to understand (“strongly regulates progenitors”?) and a bit discouraging as well since one of the virtue of the deleter line used here is to leave aside these previously investigated aspects of GSK3. They were also somewhat less interesting from a developmental neurobiology viewpoint since they dealt in essence with proliferation as opposed to migration. In a similar vein, it is unfortunate that the Abstract fails to mention that GSK does not seem to be involved in the migration of cortical interneurons. Finally, the role of GSK in the Semaphorin pathway would be worth emphasizing in the Abstract and perhaps better explained in the Introduction. The important [13] study is briefly mentioned there as well, but it would be useful if it would be better emphasize at the expense of some of the Wnt pathway discussion that could be kept for the Discussion.

---

## [Author Response]

We have thoroughly revised our manuscript based on helpful comments of the reviewers. In particular, a weakness of the prior version that diminished novelty was inadequate verification of several of the lines of mutant mice on which we reported negative findings. As instructed by the editors, we have carefully verified protein reduction after *Neurod6-Cre* mediated recombination in each of the six lines and provided quantification of cortical neuronal migration. Most of these results are shown in a new Figure 6 where new confocal images of Cux1 expressing neurons in mutants and controls are shown together with Western blots documenting strongly reduced levels of proteins in the mutants.

Quantification is shown in supplemental material linked to this new data in Figure 6—figure supplement 1.

We have also carefully commented on all of the points raised by each reviewer and revised wording in the text in accordance with reviewer suggestions.

Reviewer #1: *[…] 1) The authors demonstrate a robust and clearly-defined deficit in the migration of newborn Cux1-positive cortical neurons as soon as they are born. In contrast, they see that the deeper-layer Tbr1-positive neurons are unaffected. Since these neurons are born quite a bit earlier, are the authors confident that their driver line Nex-Cre actually causes recombination early enough to affect these earlier-born neurons? Also, most Tbr1-positive neurons are generated directly by radial glia, while the Cux1-positive neurons are largely generated* via *intermediate progenitors. Since intermediate progenitors are targeted by their Nex-Cre construct, might this provide an explanation for these differences?*

*While it is not essential for the authors to address these issues experimentally here, they should be more cautious in their interpretation of these differences*.

We are in complete agreement with the reviewer on this point. In fact we did not perform extensive analysis of the Tbr1-positive cells precisely because we did not think we could draw a strong conclusion about GSK-3 regulation of this population.

In order to make it clear that the strong GSK-3 regulation of migration may not apply to early born Tbr1-positive neurons, we modified the text in the Discussion as follows: “Deeper layer neurons expressing Tbr1 were not strongly influenced, but we cannot be certain that GSK-3 protein was fully depleted in these earlier born neurons at the time migration was occurring.”

*2) In*
Figure 2*, using mice floxed for both GSK3A and B, it looks like there are many more Cux1-positive neurons in addition to their deficit in migration. Is it possible that inducible deletion of GSK3 in intermediate progenitors causes an expansion of cortical neurons which in turn affects their migration indirectly? The authors need to address this possibility by quantifying the relative proportions of Cux1-positive neurons in these floxed mice relative to the controls*.

We think this unlikely because our cell autonomous recombination studies (Figure 4) where recombination is targeted to postmitotic neurons with *Neurod1-Cre* show that a heavy majority of cells did not migrate to the upper layers. Also time lapse imaging of cells that are clearly neurons show that migration is strikingly delayed for almost all neurons imaged (Figure 4).

In addition, as suggested by the reviewer, we quantified numbers of Cux1 positive cells in relation to total cell number in both *Gsk3* mutants and controls. The number of Cux1 neurons in mutants was very similar to the number in controls.

*3) In*
Figure 2*, the authors show statistical significance, but they only have an n=2 for controls. How* can *this be the case? I realize that some authors count each section as an n, but this isn't really valid for these types of analyses*.

N = 2 for controls and n = 3 for the experimental refers to number of animals studied. These were littermate controls obtained from 2 separate pregnant females, multiple separate coronal sections of the electroporated cortex was studied in each animal (on average, 300 neurons per section). We do not necessarily get appropriate genotypes with matched controls from every litter electroporated. Since the n is low and the effects are relatively modest, we moved the overexpression image to Figure 2—figure supplement 1 and emphasize the point that overexpression of *Gsk3* does not delay migration (based on n = 3 animals) as would have been predicted from prior studies.

*4) One of the main conclusions of this manuscript is that GSK3 is essential for radial but not tangential migration, based upon their analysis of cortical excitatory versus MGE-generated inhibitory interneurons. This conclusion isn't really warranted by the data they show. In particular, while*
Figure 3
*does show that inhibitory interneurons make it into the cortex, these data need to be quantified and it needs to be shown that GSK3 has actually been deleted in these neurons to draw this conclusion*.

We have addressed both of these important points. A Western blot of MGE lystates has been added to supplementary material, Figure 3—figure supplement 1, which demonstrates strong reduction GSK-3 protein in precursors and newly generated inhibitory neurons after recombination with *Dlx5/6-Cre*. Quantification showing only modest if any effects on migration is included in this supplement. We have also added the caveat to the text that we assessed only gross measures of tangential migration as we did not asses migration of interneuron subtypes in this study: “These results are not meant to imply that migration of interneurons was normal in every respect as we did not assess migration of specific interneuron subsets.”

*5) Similarly, the authors conclusions about direct effects of GSK3 ablation on dendritic orientation are also difficult to draw based upon their experimental design. In particular, while I agree that their data convincingly show inappropriate orientation of the dendrites of Cux1-positive neurons in layers 2-3, the authors cannot rule out the possibility that these neurons were delayed or perturbed in their migration to this location, and that the dendritic deficits are a secondary consequence of any such perturbations. The authors need to deemphasize their conclusions that these are direct effects unless they have data dissociating migration from later events such as dendritic development*.

The reviewer makes an interesting point. However, we are not in complete agreement. We specifically chose the few neurons in normal locations in Layer 2-3 for analysis. Figure 4 shows that some *Gsk3* deleted neurons are normally positioned by E19 and Figure 1—figure supplement 1 shows a few Cux 1 neurons normally positioned as early as E16. It seems fair to assume that these neurons did not undergo a prolonged migration delay, presumably because GSK-3 levels were not drastically reduced in these few neurons at these early stages. As the reviewer acknowledges, the abnormal orientation of the dendrites is striking. We think the most plausible explanation is that dendrites of *Gsk3* deleted neurons do not respond normally to directional cues that are present.

We agree that we can’t absolutely prove that dendritic polarization is directly related to GSK-3 abrogating signaling as opposed to an abnormality acquired during migration. We have therefore modified the title to: *“*GSK-3 signaling in developing cortical neurons is essential for radial migration and dendritic orientation” which is certainly true and avoids the terms “regulate” and “polarization”. We have also modified our text to reflect reviewer concerns on this point as follows: A) We changed the section heading from: “GSK-3 regulates dendritic orientation” to “Abnormal dendritic orientation in *Gsk3* mutants”

B) We added text: “Images at E16 (Figure 1—figure supplement 1) and E19 (Figure 4’) show that a few *Gsk3* deleted neurons are normally positioned and suggest that these normally positioned neurons did not undergo a substantial delay in migration.”

*6) Data in*
Figures 6 and 7
*showing negative results vis a vis migration with multiple signaling proteins are a nice addition to the manuscript. However, to really substantiate these negative data, it would be important to (i) show that the Cre-mediated recombination really did occur as predicted, and (ii) show some kind of quantification to rule out the possibility that there are indeed neuronal deficits, but that they are just more subtle than with GSK3 knockout*.

Again, a very helpful point. As noted above, we have carefully verified protein reduction and provided quantification of migration in five of the six lines where we report negative results, Figure 6 and Figure 6—figure supplement 1. For comparison we show quantification of the strikingly abnormal quantification in *Gsk3* mutants in Figure 2, new panels A and C. For *Dvl123:Neurod6* we did not obtain enough 6 allele mutant mice for both Westerns and detailed quantification. We do show a representative image based on inspection of multiple sections from n = 3 of the 6 allele mutant mice. We hope we can keep the image in the paper as it will be of substantial interest to researchers in this field.

*7) I'm not sure the small amount of microarray data shown in*
Figure 6
*really adds anything to the manuscript. I would either add all of the microarray data or remove this*.

We have now removed the Table shown in former Figure 6 and simply mention results in the text. The full array has been deposited in GEO, NCBI for reference (GEO accession number GSE58727) and will be released to the public shortly.

*8) The data on Crmp-2 are important and interesting. The Dcx data are potentially also very interesting, but the authors don't show that there is no change in total Dcx levels. In the absence of such data, it is difficult to know whether relative phosphorylation of Dcx is actually decreased*.

We probed for total DCX in these blots as directed. The reduction in pDCX remains very striking. The new results are shown in a revised Figure 7.

Reviewer #2: *[…] The data are mostly well-documented and convincing. Indeed, as authors claim, to my knowledge, this is the first genetic demonstration that GSK-3's are required for specific aspects of neuronal migration. My main concern is with the novelty of the findings, relative to the J Nsci study by Asada and Sanada which also showed a requirement for GSK-3 in migration, albeit through an acute, non-genetic manipulation. Other studies, including Shi et al. Curr Biol 14, 2025-2032 (2004) have also implicated GSK-3 in dendrite vs. axon polarity*.

We addressed this point in an exchange with the editors: our findings are in direct conflict with the Asada study. We did *not* verify that *inhibition* of GSK-3 via an LKB1→GSK-3→APC (dephosphorylation) signaling cascade is required for neuronal migration as was published in that paper. Analysis of GSK-3 in the Asada paper relied exclusively on activation of GSK-3 via the GSK-3β S9A construct and the demonstrated effects on migration were transient. We understand that GSK-3 inhibition due to ser9/ser21 phosphorylation and subsequent dephosphorylation of key cytoskeletal regulators has previously been implicated in vitro in elaboration of axons and migration of non-neuronal cells. However, we and others (Courchet et al. Cell 2013) do not find evidence that LKB1 signaling regulates migration in vivo*.* Finally, published work that we cite on GSK-3 mutant knock-ins that are viable and exhibit grossly normal brain development, suggest it is unlikely that ser9/ser21 phosphorylation of GSK-3 strongly regulates cortical neuron migration in vivo*.* Rather, our study shows that GSK-3 activity is required for cortical neuronal migration in vivo. Our evidence suggests that GSK-3 phosphorylation of key downstream cytoskeletal regulators (rather than dephosphorylation) is crucial for migration.

Additionally, while I appreciate the lack of migration phenotypes in other KOs examined here, contrasted with Asada's manipulations of LKB1, I am concerned about the extent of the loss of function induced. The fact that in the b-cat KO, the authors fail to detect any effect on Wnt signalling targets makes me wary of their strong conclusions. Especially, since the effect of KO on b-catenin expression is barely more than a two-fold change. How do we know that the induction of those KOs is efficient enough to warrant the strong conclusion that LKB1, Pten, b-catenin etc, are not required for migration?

To clarify, we did not assess Wnt target genes in *Ctnnb1*^*Ex3*^*:Neurod6* mice. Rather, we assessed Wnt target genes in *Gsk3:Neurod6* mice. Reviewer 1 felt that this aspect should be given less emphasis and we have changed the presentation of the data as directed (see response to point 7 above)

We now show strikingly reduced protein levels after *Neurod6-Cre* mediated recombination for all of signaling mediators studied (new Figure 6—figure supplement 1) as noted in point 6 above. These strong protein reductions demonstrate that *Neurod6-Cre* was highly effective in generating recombination of all of the floxed alleles used in this study.

Reviewer 3: *[…] My main recommendation would be to revise the Abstract to better highlight the interest of the study. In particular, the first sentence is difficult to understand (“strongly regulates progenitors”?) and a bit discouraging as well since one of the virtue of the deleter line used here is to leave aside these previously investigated aspects of GSK3. They were also somewhat less interesting from a developmental neurobiology viewpoint since they dealt in essence with proliferation as opposed to migration. We have revised the Abstract and removed the first sentence as recommended*.

*In a similar vein, it is unfortunate that the Abstract fails to mention that GSK does not seem to be involved in the migration of cortical interneurons*.

We have added a new sentence to the Abstract: “In contrast, tangential migration is not grossly impaired after *Gsk3* deletion in interneuron precursors”. This point is also now mentioned in the Introduction.

*Finally, the role of GSK in the Semaphorin pathway would be worth emphasizing in the Abstract and perhaps better explained in the Introduction. The important*
[13]
*study is briefly mentioned there as well, but it would be useful if it would be better emphasize at the expense of some of the Wnt pathway discussion that could be kept for the Discussion*.

Because of the word limitations of the Abstract, we cannot emphasize the Sema pathway here. However, we have revised the Introduction to include information on the Sema pathway as instructed.